



# Analysis of causes of decreasing inflow to the Lake Chad due to climate variability and human activities

Rashid Mahmood, Shaofeng JIA

Key Laboratory of Water Cycle and Related Land Surface Processes/Institute of Geographic Science and Natural Resources
Research, Chinese Academy of Sciences, Beijing 100101, China

*Correspondence to*: Shaofeng JIA, Rashid Mahmood, (jiasf@igsnrr.ac.cn, rashi1254@gmail.com, )

**Abstract.** In the 1960s, Lake Chad was the world's sixth largest water body, which has since shrunk dramatically from a surface area of 25,000 km$^2$ to only 2,000 km$^2$ in the following 40 years. In the present study, hydro-climatic variability in the Chari-Logone, Komadugu-Yobe, YENG (Yedseram, El-Beid, Ngadda and Gubio basins) as well as Lake Fitri basins and

decreasing streamflow to Lake Chad due to climate variability and human activities were separated and quantified using trend analysis, change point analysis, and hydrological approach, for the period of 1951–2015. The results showed very strong signals ($\alpha=0.001$) of increasing trend in mean temperature, with an average increase of 1.4 °C, and very weak ($\alpha=0.1$) to strong ($\alpha=0.01$) decreasing signals in precipitation, with an average decrease of 15%. In case of streamflow to Lake Chad, very strong decreasing trends were observed, showing 67% reduction for the whole period. The north-eastern parts were most affected

parts in case of increasing temperature and decreasing precipitation. Decreasing flow due to both climate variability and human activities were ranged from 34% to 45% in different decades, from 1972 to 2013. On the whole, a 66% of total decline in streamflow was observed due to human activities and 34% due to climate variability. Most reduction in streamflow (59%) due to climate variability was explored only during 1982–1991 because a devastating drought was occurred during this period. Since human activities caused most reduction in streamflow to Lake Chad than climate, inflow to the lake can be improved by

reducing or properly managing the human activities and using sustainable water resources management.

## 1 Introduction

Lake Chad (LC), one of the biggest lakes in the world, is located in the Lake Chad basin (LCB), the largest endorheic basin in Africa (Gao et al., 2011). LC straddles the borders of Chad, Nigeria, Niger, and Cameroon, as shown in Fig. 1. As of 2011, it provides livelihoods to more than $30 \times 10^6$ people across the basin. It is a vital source of freshwater and fishing and also

provides water for pastoral and agricultural land. In the 1960s, LC was the world's sixth largest water body, with a surface area of 25,000 km$^2$. However, in the subsequent 40 years, it has dramatically reduced to 2,000 km$^2$ (LCBC, 2011;Coe and Foley, 2001) and even about 300 km$^2$ in the 1980s (Gao et al., 2011). In 1975, the lake was divided into two parts (i.e., the northern pool and the southern pool) because of devastating drought over the African Sahel belt. Since then, the northern pool has been rarely and partly inundated (Lemoalle et al., 2012). According to  Zhu et al. (Zhu et al., 2017), the southern pool of





the lake has an average annual surface area of 1,446 km$^2$ for the period of 1992–2016. Same like other lakes that situated in closed drainage basins, the variation of water level in LC mostly depends on the streamflow from different rivers in the LCB, which varies according to the annual rainfall in the basin (Coe and Foley, 2001;Lemoalle et al., 2012). Due to shallow water level (even less than 7 m), LC is very sensitive to temperature, precipitation, and runoff changes (Lemoalle et al., 2012).

LC is mostly fed by two river systems: the Chari-Logone River system and the Komadugu-Yobe River System (Singh et al., 2006;Gao et al., 2011). The Chari-Logone River (CLR) flows in the south-eastern parts of the LCB  and enters to the southern pool of the lake (Fig. 1), contributing more than 90% (27.14 km$^3$ yr$^{-1}$, mean for1960–2013) to the lake, and the Komadugu-Yobe River (KYR) runs in the western parts of the LCB and enters to the northern pool of the lake (Fig. 1), contributing  2–5% (0.56 km$^3$ yr$^{-1}$, mean for 1961–2013) (Gao et al., 2011;Lemoalle and Magrin, 2014). In addition, the Yedseram, El-Beid,

Ngadda, and Gubio (YENG) Rivers enters from the southwest of LC (Fig. 1) and contributes about 1–2%. However, the Batha River to the east of Lake Chad (Fig. 1) does not directly contribute to Lake Chad but to Lake Fitri (Komble et al., 2016). Agriculture is the main water user in the basin, which provides the major livelihood to about 60% people of the LCB. During 1983–1994, the water demand in the agriculture sector was increased by twofold because of rapidly increasing population in the LCB from 1960 ($13 \times 10^6$) to 1990 ($26 \times 10^6$). At the present, an area of about 135,000 hectares is irrigated in the basin.

The most devastating droughts in the 1980s extremely reduced the water level in LC, resulting in the shortage of water for irrigation canals. Moreover, building dams have greatly affected the river flow to the lake. About 23 earth dams have been constructed in Kano and Hadejia basins, resulting reduction in the KYR's flow to the lake from 7 km$^3$ yr$^{-1}$ to 0.45 km$^3$ yr$^{-1}$ due to impoundment of reservoirs in Kano province (Singh et al., 2006).

During the last few decades, dramatic rise in the concentration of greenhouse gases has resulted in global warming and global

energy imbalance (Mahmood and Babel, 2013). Consequently, the global average temperature has increased by 0.85 °C during 1880–2012 and by 0.74 °C ± 0.18 °C only in the last century (906–2005) (IPCC, 2013). This global warming can greatly disturb the hydrologic cycle of world, resulting changes in the availability of water and changes in extreme events such as flood and droughts (Khattak et al., 2011). The disturbance of hydrological cycle can seriously affect water demand in different sectors, public health, energy exploitation, ecosystem, and food security. However, the impacts of climate change on

hydrologic system may vary from region to region (Chu et al., 2010). Recently, climatic variability is a great challenge to the world and especially to Africa. Since the most parts of Africa such as Niger, Central African Republic etc. are not economically strong enough to cope with changing climate and climatic variability, this rapidly changing climate can greatly affect the water resources of Africa. One noticeable example is the LCB, where inflow to the lake has decreased by 70–80% and surface area of the lake about 90% over the last 40 years (Coe and Foley, 2001;Gao et al., 2011). Moreover, the lake was separated into the

northern and the southern pools during 1973–195 because of devastating decrease in precipitation (Lemoalle et al., 2012). Since 1970, water level in the lake has declined greatly because of climate variability and human activities, and the discharge of the CLR (contributes more than 90%) has decreased by 75% only over the last 40 years, which is an alarming situation in the basin (Jun and Bantin, 2017;Singh et al., 2006). According to (Coe and Foley, 2001), climate variability and water withdrawal in the basin are the main causes of fluctuation of water level in Lake Chad. They found that climate variability and



human activities equally caused reduction of water level in the lake during 1983–1994. However, that estimation was based on hypothetical irrigation water use and carried out 17 years ago. So it is essential to conduct a newest and more precise study to assess the climate variability in the most active parts of the basin, also known as conventional basin, and to separate and quantify the impacts of climate change and human activities (e.g., irrigation water withdrawal), which can help to cope with

climate change, to manage the impacts of human activities on the water resources of the basin, and to improve the water resources management practices in the basin.

There have been some new literatures to explore the climate variability in some parts of the LCB: (Nkiaka et al., 2017;Okonkwo et al., 2014;Adeyeri OE et al., 2017;Funk et al., 2012). Nkiaka et al. (2017) used trend analysis to explore the precipitation variability (1950–2000) in the Logone River basin. They used only 11 rain gauges, which are unevenly distributed

in the Logone basin. Okonkwo et al. (2014) conducted a study to link ENSO and precipitation with the water level fluctuation in LC, using precipitation amount (only June–September) from CRU for the period of 1970–2010. However, they used only weighted mean of the northern part and the southern part of the LCB, dividing the whole basin from the great barrier of the lake. Adeyeri OE et al. (2017) studied precipitation trends in the Komadugu-Yobe River basin using some observed and gridded data of Princeton University for the period of 1979–2015. Funk et al. (2012) conducted a study to explore temperature

and precipitation variability in Chad. They used only 13 temperature gauges and 117 precipitation gauges for the whole Chad for the period of 1900–2009. However, they conducted study for only rainy season (June–September). Each of the above studies cover some small parts of the LCB and no studies are reported covering the whole conventional basin—the most active part of the LCB entirely contributes to LC—and even Chari-Logone River basin, which contributes more than 90% water to LC. In addition, most of the studies concentrated on the precipitation variability.

Coe and Foley (2011) studied to find out the impacts of climate variability and human water use on streamflow to LC from the CLR only for one decade (1985–1994) with respect to 1956–1975. They calculated the changes in stream flow due to climate and human activities by hydrological model run with and without hypothetical irrigation withdrawal. Similar kind of study was reported by Gao et al. (2011) to explore the causes of shrinking LC for the period of 1952–2006, but their focus was to find out the causes of split of LC and how to recover to its size of 1960s. Many other studies, such as (Lemoalle et al.,

2012;Buma et al., 2016;Zhu et al., 2017;Leblanc et al., 2007;Ndehedehe et al., 2016) have been reported in the basin to explore the lake water level changes, changes in terrestrial water storage, and groundwater changes. However, most of these studies concentrated on the LC's area. No studies have been reported to cover the most active parts, completely, of the LCB.

Thus, in the present study, 1) hydro-climatic variability was assessed in the most active parts of the LCB (i.e., Chari, Logone, Komadugu, Yobe, Yedseram, El-Beid basin, Ngadda, Gubio, and Batha River basins), for the period of 1951–2015, which is

the mandate of Lake Chad Basin Commission; 2) the impacts of climate variability and human activities on the streamflow to Lake Chad were separated and quantified in the Chari-Logone River basin because it contributes more than 90% water to LC. We used a different hydrological approach than applied by Coe and Foley (2011) and for a longer data period (1951–2015). We also modified the equation which is used for calculating the impacts of climate variability on streamflow (more detail in methodology). Due to limited and scarce observed data, the gridded monthly meteorological data of CRU and monthly





observed streamflow data were used in this study for the period of 1951–2015, which was converted into daily form as requirements of hydrological model, using MODAWEC (Monthly to Daily Weather Convertor) model.

This study is a part of an on-going project "Inter Basin Water Transfer Project (IBWTP) Feasibility" of PowerChina International Limited. The main focus of the project is study the feasibility of restoring rapidly shrinking Lake Chad to its

normal ecological level by transferring water from the Congo River. The present study will help to understand hydro-climatic changes in the LCB and to understand the impacts of climate variability and human activity on the inflow to Lake Chad separately. This study will help in choosing better adaptation options to cope with climate variability and water resources management practices to deal with human impacts.

## 2 Study area

The Lake Chad basin (LCB), the world's largest closed drainage basin covering an area of $2.5 \times 10^6$ km² (about 8% of Africa) (Gao et al., 2011;Coe and Foley, 2001),  is situated between 5.19–25.29°N latitude and 6.85–24.45°E longitude, as show in Fig. 1. The LCB shares the borders with Algeria, Cameroon, Central African Republic, Chad, Libya, Niger, Nigeria, and Sudan. The most area of the LCB, 44%, is located in Chad followed by Niger, 29%. The LCB receives an average annual rainfall of 415 mm, ranging from 1215–1600 mm in the south-western parts of the basin (Central African Republic) to 20–150

mm in the northern parts such as Algeria. The most active parts of the LCB that almost entirely contributes water to LC is known as conventional basin (Fig. 1). This is the mandate of Lake Chad Basin Commission (LCBC), established in 1964 and composed of four countries: Cameroon, Chad, Niger, and Nigeria. At that time, it was about 20% (427 300 km²) of the LCB's area (Frenken, 1997;Lemoalle and Magrin, 2014;Odada et al., 2005). In 2012, it was extended to an area of 967 000 km², which is around 40% of the LCB's area (Komble et al., 2016). The most active parts of the conventional basin includes the

following river basins: the Chari, the Logone, the Komadugu, the Yobe, the Yedseram, the El-Beid, the Ngadda, the Gubio, and the Batha River basins (Lemoalle and Magrin, 2014), as shown in Fig. 1. These rivers mostly contribute to LC, especially the Chari-Logone River and the Komadugu-Yobe River. Lake Fitri located to the east of LC is fed entirely by Batha River (Fig. 1). Hereafter, YENG will be used for the Yedseram, the El-Beid, the Ngadda, and the Gubio River basins. In the present study, the Chari-Logone River basin (CLRB), the Komadogu-Yobe River basin (KYRB), the YENG, and Lake Fitri basin

(LFB), which cover the most parts of the conventional basin (ConvB), as shown in Fig. 1, were selected for the assessment of hydro-climatic variability. However, hydrological modelling was performed only in the CLRB to separate and quantify the streamflow changes in Lake Chad due to climate variability and human activities because this river is the main source of water to Lake Chad, contributing more than 90%. Some basic characteristics of these basins are given in Table 1.

### 2.1 Hydro-climatic condition in the study area

Table 1 shows observed mean annual inflow to Lake Chad for the period of 1951–2013 as well as mean annual temperature and precipitation in the LCB, the ConvB, the CLRB, the KYRB, the YENG, and the LFB for the period of 1951–2015,





calculated from CRU climatic data. The LCB receives an annual precipitation 465 mm and has 26.7 °C as an average annual temperature. The CLRB is the wettest basin, with mean annual precipitation of 935 mm, and the LFB is the driest part, with a mean precipitation of 527 mm. Precipitation in the study area not only varies geographically but also seasonally. Most of the precipitation (80–90%) occurs only from June to September. In contrast, the LFB and the CLRB are the hottest and the coldest parts, with mean temperature of 28.3 °C and 26.7 °C, respectively. The CLR contributes an annual flow of 931 $m^3 s^{-1}$ (92% of total flow) to LC, for the period of 1951–2013, and the KYRB contributes about 74 $m^3 s^{-1}$ (about 5%). Other rivers contribute even less than 3% to LC, as described in Table 1.

## 3 Data description

### 3.1 Station hydro-meteorological data

Historical observed time series play the central role in studies like detecting climatic changes, streamflow changes, lake level changes, water resources management, and water resources planning. Monthly data of 11 meteorological (six for the period of 1950–2013 and other for1985–2013) stations and 7 hydrometric stations (four for 1997–2007, two for 1951–2007, and one for 1951–2013) were collected from the Lake Chad Basin Commission (LCBC), as shown in Fig. 2a, which shows a serious scarcity of climatic data in the LCB. The inflows to LC from the CLR, the KYR, and the El-Beid Rivers were collected for the period of 1951–2013. Except N'djamena discharge gauge, other gauges either have short data period or of high missing values. Although, collected meteorological stations have long time series of precipitation and less missing values, the numbers of stations are seriously scarce, not even covering the main basin (the CLRB) and unevenly distributed in the basin.

### 3.2 Gridded meteorological data

Recently, globally interpolated gridded observed datasets (e.g., Climate Research Unit (CRU) and Global Precipitation Climatology Centre (GPCC)), satellite (e.g., CICS High-resolution Optimally-interpolated Microwave Precipitation from Satellite (CHOMPS)), reanalysis (e.g., National Centre for Environmental Protection (NCEP)), and combination of satellite and observed (e.g., Tropical Rainfall Measurement Mission (TRMM)) have been created by the world leading scientific research centres for understanding and prediction of weather, water, and climate of the world.

Due to lack of high quality and long term historical observations in the basin, in the present study, monthly precipitation (PP), mean temperature at 2 m height (TM), maximum temperature (TX), minimum temperature (TN), wet days (WD), and potential evapotranspiration (PE) were obtained from the latest version (i.e., CRU-TS4.00) of the CRU dataset for the period of 1951–2015, and the extracted CRU data for each sub-basin are shown in Fig. 2a. This is a high resolution monthly gridded dataset of 0.5° × 0.5° grid resolution, from 1901 to 2016, containing the following variables: PP, TM, TX, TN, WD, PE, diurnal temperature range, vapour pressure, cloud cover, and frost-day frequency (Harris et al., 2014).This dataset used in many studies for the evaluation of simulated data from global climate models, regional climate modes, and hydrological models (Chiyuan et al., 2014;Smiatek et al., 2009;McMahon et al., 2015), where the observed point data is not of good quality and scarce.



### 3.3 Land cover, soil, and elevation data

Land cover can have great effect on hydrologic processes of a basin, especially the density of plant cover and the morphology of plant species can greatly influence these processes (Ghoraba, 2015). The land cover data for the LCB was derived from the U.S. Geological Survey (USGS) (https://earthexplorer.usgs.gov), with 1 km spatial resolution. The soil data of 1 km resolution was acquired from the Harmonized World Soil Database (HWSD), which  was established by FAO with the collaboration of  the International Soil Reference and Information Centre, the International Institute Of Applied Systems Analysis, the World Soil Information, the Institute of Soil Science, and the Joint Research Centre of the European Commission (http://www.fao.org/soils-portal/soil-survey/soil-maps-and-databases/harmonized-world-soil-database-v12/en/).      Basin's elevation information is one of the primary sources to extract topographic information of a terrain, especially in hydrological studies. Recently, a digital elevation model (DEM) is the most common form of elevation data   (Singh et al., 2015). In this study, a DEM of NASA's Shuttle Radar Topography Mission (SRTM) was obtained from the USGS (https://earthexplorer.usgs.gov). SRTM-DEM covers most part of the world, with a spatial resolution of 3 arc-seconds for global coverage. The DEM, the land cover types, and the soil types for the LCB are shown in Fig. 2b, c, and d, respectively.

## 4 Methodology

### 4.1 Input data preparation

Since the study area (911 000 km$^2$) is quite large, the whole study area was divided into 84 sub-basins (37, 15, 17, and 15 sub-basins for the CLRB, the KYRB, the YENG, and the LFB, respectively) using DEM, as shown in Fig. 2a, which was processed by HEC-GeoHMS on ArcGIS platform. In addition, other basin characteristics such as rivers' slops, sub-basins' areas, river lengths, longest flow paths, basins' average elevations, and streamlines were also extracted during this process, which were required for hydrological modelling. This information along with the information from soil and land cover datasets were used to extract the initial estimations of hydrological properties of the basin, such as time of concentration, storage coefficient, maximum soil moisture deficit, initial deficit, infiltration rate, percentage of imperviousness etc. However, the exact estimates about these parameters were obtained during the calibration process.

Since monthly collected observed data was extremely scarce in the basin, meteorological data (i.e., TM, TX, TN, PP, PE, and WD) was obtained from CRU climatic dataset, which is also based on monthly interval and gridded based. However, HEC-HMS requires point data for semi-distributed modelling. For each sub-basin, meteorological variables were obtained by taking the average of all CRU grids covering that sub-basin. In this way, a total of 84 time series (37, 15, 17, and 15 time series for the CLRB, the KYRB, the YENG, and the LFB, respectively) for each variable (i.e., TM, TX, TN, PP, PE, and WD) were constructed in the LCB.

In addition, HEC-HMS also requires daily or sub daily (ranging from 1 minute to 1day) meteorological inputs to simulate flow. Therefore, CRU monthly TX, TN, and PP were converted into daily time series using the MODAWEC (MOnthly to



DAily WEather Convertor) model (Liu et al., 2009). MODAWEC converts monthly PP, TX, and TN to daily values while it preserves the monthly total PP and monthly average TX and TN. The required inputs for the model are monthly PP, TX, TN, and WD, and outputs are daily PP, TX, and TN. Since PP is used as driving variable in MODAWEC, it is considered as a parametric weather generator. It generates the amount and occurrence of PP independently, and TX and TN are produced based

on the generated daily PP. The methods used in this model for the generation of daily PP and temperature are explained by Liu et al. (2009). The monthly streamflow at N'Djamena station was simply converted into daily data by taking monthly mean values for all days of corresponding month because in the present study, the main focus was monthly and annual flow time series.

## 4.2 Trend Analysis

Many parametric and non-parametric methods have been developed to detect trends and to estimate the magnitude of trends, as reviewed in (Esterby, 1996;Zhang et al., 2006;Sonali and Nagesh Kumar, 2013). Parametric tests are considered to be more powerful, but time series must be normally distributed. As hydro-climatic times series, especially PP, do not pass normality test, therefore, non-parametric methods are commonly used for trend detection in time series, as in (Burn et al., 2004;Fu et al., 2010;Wang et al., 2012;Tekleab et al., 2013;Oyerinde et al., 2014), and are considered to be more robust relative to parametric

methods (Hess et al., 2001;Khattak et al., 2011;Zhang et al., 2006;Sonali and Nagesh Kumar, 2013). Thus, a non-parametric method, Mann-Kendal (MK) (Mann, 1945;Kendall, 1975), was applied in present study to identify the hydro-meteorological trends in the LCB. For a time series $x_1, x_2, x_3, ..., x_n$, with $n > 10$, the following equation is used to calculate the MK test statistic (Z), as in (Hu et al., 2015;Mahmood and Jia, 2017;Feng et al., 2016):

$$Z = \begin{bmatrix} \frac{S-1}{\sqrt{V(S)}} & \text{for } S > 0 \\ 0 & \text{for } S = 0 \\ \frac{S+1}{\sqrt{V(S)}} & \text{for } S < 0 \end{bmatrix} \qquad 1$$

Where S and V(S) are calculated as below:

$$S = \sum_{i=1}^{n-1} \sum_{j=i+1}^{n} sgn(x_j - x_i) \qquad 2$$

in which

$$sgn(x_j - x_i) = \begin{bmatrix} 1 & \text{for } (x_j - x_i) > 0 \\ 0 & \text{for } (x_j - x_i) = 0 \\ -1 & \text{for } (x_j - x_i) < 0 \end{bmatrix} \qquad 3$$

$$V(S) = \frac{1}{18} \left[ n(n-1)(2n+5) - \sum_{p=1}^{q} t_p(t_p - 1)(2t_p + 5) \right] \qquad 4$$





Where $q$ is the number of total tied groups (a tied group contains a set of same values in a dataset) and $t_p$ is the number of

values in each tied group. Positive $Z$ values are the indication of upward trends in time series, and the opposite show downward

trends. Trends (upward or downward) are then checked that either trends are statistically significant or not. For example, if the

absolute value of $Z$ is greater than $Z_{1-\alpha}$, (e.g., $Z_{1-\alpha}$ calculated at α = 0.05); the null hypothesis (i.e., no trend) is rejected, and

alternative hypothesis (i.e., significant trend) is accepted at α = 0.05. The $Z_{1-\alpha}$, is a critical value used to decide either the time

series has significant trend or not.

To determine the magnitude of hydro-climatic trends, the Sen's slope method (Sen, 1968) is frequently applied in hydro-

climatic studies, as in Khattak et al (2011), Burn et al (2004), Kumar et al. (2009), and Mahmood and JIA (2017). Since this

method is robust against outliers, it effectively quantifies a trend in a time series. Sen's slope ($b$) is calculated as below:

$$b = median \left[ \frac{x_j - x_i}{j - i} \right] \quad \text{for all } i < j \qquad 5$$

Where $x_j$ and $xi$ are the data points at time $j$ and $i$, respectively. Before applying the MK test, hydro-climatic time series must

be free of serial correlation because serial correlation can mislead the actual result of trends. Thus, in the present study, trend-

free prewhitening (TFPW) method developed by Yue et al. (Yue et al., 2002) was used to remove the serial correlation, if

existed, from the hydro-meteorological time series. This method has frequently been used to remove the serial correlation

effects from the time series, as in (Khattak et al., 2011;Kumar et al., 2009;Burn et al., 2004). In the present study, trends were

detected on annual time series of temperature and precipitation in the CLRB, the KYRB, the YENG, and the LFB for the

period of 1951–2015, on four significance levels (i.e., α = 0.10, 0.05, 0.01, and 0.001). For clarification, these basins were

divided into 3–5 regions (e.g., northern, southern, eastern, and western parts).

## 4.3 Change point detection

Before starting hydrologic modelling and assessing the impacts of climate variability and human activities, there was essential

to find out a stable period (baseline period) where the impacts of human activities were considered to be minimum. In the

present study, we used two statistical tests to detect the change point year in the hydro-climatic time series in the CLRB:

Standard Normal Homogeneity Test (Alexandersson, 1986) and Worsley Likelihood Ratio Test (Worsley, 1979).

### 4.3.1 Standard Normal Homogeneity Test

Standard Normal Homogeneity Test (SNHT) developed by Alexandersson (1986) has been used frequently by the researcher

for the detection of change point years (break points) and for inhomogeneity check in hydro-climatic time series, as in (Jaiswal

et al., 2015;Vezzoli et al., 2012;Štěpánek, 2009;Freiwan and Kadioğlu, 2008;Mahmood and Jia, 2016;Göktürk et al., 2008).

For a time series like $y_1, y_2,...,y_n$ with $n$ observations, the SNHT can be formulated as below:





$$T(k) = kz_1^2 + (n-k)z_2^2 \quad for \ k = 1, 2, \dots, n \qquad 6$$

where

$$z_1 = \frac{1}{k} \sum_{i=1}^{k} \frac{y_i - \bar{y}}{\sigma_y} \quad and \quad z_2 = \frac{1}{n-k} \sum_{i=k+1}^{n} \frac{y_i - \bar{y}}{\sigma_y}$$

$$T_{max} = \max_{1 \le k \le n} T(k)$$

Test statistic $T(k)$ compares the mean of the first $k$ values with the mean of the last $n$-$k$ values. The highest value ($T_{max}$) obtained from $T(k)$ statistics is the indication of the change point in the time series. To check the change point statistically significance, $T_{max}$ is compared with the critical values (given in Table III of (Wijngaard et al., 2003)) at specified significance

10 level (e.g., α=0.05). $\sigma_y$ and $\bar{y}$ are standard deviations and mean values of time series.

### 4.3.2 Worsley Likelihood Ration test

Worsley Likelihood Ration test (WLRT) developed by (Worsley, 1979) explores the most likely position of a change in mean value of a time series ($y_1, y_2, \dots, y_n$). In this method, deviations from the mean are summed up at each point ($k=1, 2, \dots, n$), and then weights are given to each sum according to their positions in the time series. This method is used frequently to explore

15 the position of jumps in time series, as in (Buishand, 1982; Vezzoli et al., 2012; Dollar et al., 2006). The formulation of this test explained by (Buishand, 1982) is given below:

$$Z_k^* = \frac{S_k^*}{(k(n-k))^{0.5}} \qquad 7$$

where

$$S_k^* = \sum_{i=1}^{k} (y_i - \bar{y})$$

$$V = \max_{1 \le k \le n} |Z_k^{**}|$$

$$W = \frac{V(n-2)^{0.5}}{(1-V^2)^{0.5}}$$

where $S_k^*$ are the sums of deviations from mean, $Z_k^*$ are the Worsley test statistics (weighted adjusted partial sums), $Z_k^{**}$ is the weighted rescaled adjusted partial sums and is obtained by dividing the sample standard deviation. $V$ is the highest value of

25 test statistic showing the position of jump (change point) in a time series. If we need to know only the position of the change point, then the calculation of $W$ (Worsley likelihood ratio) is not necessary. However, if we need to check the level of significance of change point, then $W$ is essential. $W$ is compared with the critical values at different significance levels to explore that either $W$ is statistical significant or not.





These tests were applied on annual streamflow at N'djamena gauge for the period 1951–2013 and also applied on annual TX, TN, TM, and PP in the CLRB for the period of 1951–2015, though not necessary for climatic variables for a hydrological approach used for assessing the changes in streamflow due to climate and human. The obtained results were also confirmed with information available about abrupt changes in streamflow in the literature.

**4.4 Description of HEC-HMS**

Hydrologic Engineering Centre-Hydrological Modelling System (HEC-HMS) is a rainfall-runoff simulation model, created by U.S. Army Corps of Engineers at Hydrologic Engineering Centre (HEC). The model is designed to solve widespread possible hydrological problems (e.g., flood hydrology, large basin water supply, and small to large urban or natural watershed runoff ) in a wide range of geographic regions (Halwatura and Najim, 2013;William et al., 2010). This modelling system is composed of four major components: 1) A mathematical model to compute streamflow in a watershed and to rout it through channels 2); a data storing and managing system (i.e., HEC-DSS); 3) a GUI (graphic user interface) to display hydrological component of a basin such as sub-basins, reaches, reservoirs, and junctions; and 4) an interface for displaying/reporting outputs in the form of tables and figures to  analyse the results (Halwatura and Najim, 2013).

HEC-HMS includes twelve loss methods (e.g., SCS curve number, initial and constant, deficit-constant, soil moisture accounting, and gridded loss method) to calculate excess PP from some simple to complex infiltration and evapotranspiration environments. Some methods are designed for continuous simulation like deficit-constant method and 5-layes soil moisture counting, while others for event based simulation like initial and constant and SCS curve number. Among these loss methods, the initial and constant loss method is a simple form of loss methods, which requires only few parameters, and the 5-layes soil moisture counting is a complex and more advance form of loss methods, but it requires high number of parameters (Mahmood et al., 2016;William et al., 2010).

For the estimation of direct runoff, HEC-HMS comprises of 7 transformation methods such as SCS unit hydrograph, Clark' unit hydrograph, and Snyder's unit hydrograph. Some of them are complex and require more input parameters, that most of the time are not easy to estimate, especially for ungauged basins. The model consists of 5 base flow methods such as recession method and 6 channel routing methods such as Muskingum for base flow estimation and streamflow channel routing, respectively. To analyse meteorological data for each sub-basin, HEC-HMS has six different kinds of meteorological models such as Thiessen polygon and inverse distance methods. However, only Temperature Index  is included in the model for calculating runoff from snowfall (William et al., 2010;Feldman, 2000).

A complete basin model setup for rainfall-runoff processes comprises of a basin model, a meteorological model, a control specification, and input time series. (Verma et al., 2010). A comprehensive description about the model formulation and various processes is given in User's Manual and Technical Reference Manual (William et al., 2010;Feldman, 2000).



## 4.5 Model setup for the study area

HEC-HMS has been used throughout the world for flood modelling, water resource assessment, impacts assessment, urban flooding, flood warning system, stream restoration, water availability, flow forecasting etc. as in (Zema et al., 2017;Halwatura and Najim, 2013;Mahmood and Jia, 2017;Yimer et al., 2009;Meenu et al., 2012;Verma et al., 2010;Ramly and Tahir, 2016).

In the present study, we included the following methods for a basin model development: the deficit and constant loss (DCL) method, the Clark unit hydrograph (CUH), Muskingum for channel routing, and constant monthly base (MCB) method. For meteorological model setup, Thiessen polygon method was used to calculate PP for each sub-basin on spatial scale. The similar kind of basin model setup has been applied in (García et al., 2008;Meenu et al., 2012;Verma et al., 2010;Halwatura and Najim, 2013).

DCL is a single layer continuous method to calculate the continuous changes in soil moisture content and consequently, provides excess PP for a watershed. The potential evapotranspiration obtained from meteorological model is used to dry out the soil layer between two rainfall events. It recovers the initial losses after a long period of no PP. There are four parameters (i.e., maximum deficit, initial deficit, constant rate, and impervious percentage) which are initially estimated using soil and land cover data as initial inputs to model but are finalized during calibration process. CLU transforms excess PP calculated

from DCL into direct surface runoff. It is a synthetic unit hydrograph method and has two parameters (storage coefficient and time of concentration) to optimize during calibration. Lastly, the flow is transferred from one point to other using Muskingum method, which is a simple mass conservation scheme for routing flow through the channels. Travel time (K) and Muskingum coefficient (X) are obtained during calibration process (William et al., 2010).

### 4.5.1 Model calibration and validation

Model calibration is a procedure to adjusted model parameters in such a way that the simulated flow captures the variations of the observed flow (García et al., 2008). Before calibration of Hydrologic model, the whole hydro-climatic data was divided into baseline period and impacted period. A baseline period was the period where impacts of human activities on streamflow were consider to be minimum in the basin. The baseline period was explored by change point analysis using two statistical tests: the WLRT and the SNHT. These tests were applied on hydro-climatic time series in the Chari-Logone basin for the

period of 1951–2015. The results showed 1971 as change point year in the streamflow time series (Table 4). So hydro-climatic data was divided into two parts: 1951–1971(baseline) and 1972–2013 (impacted). In the present study, the model was calibrated and validated for the baseline period to produced natural flow for the whole period (1951–2013) because we had to access the changes in streamflow due to climate variability and human activities separately. So a data period of 10 years from 1956 to 1965 was used for calibration and two periods, i.e., 1951–1955 and 1966–1971, for validation. The basin

characteristics, i.e., soil properties and land covers were assumed to be constant throughout the simulation period. Univariate gradient optimization method was applied to minimize the objective function (peak weighted root mean square) during optimization of model parameters.



In the present study, the simulated flow was compared with observed data using four commonly used statistical indicators, i.e., coefficient of determination ($R^2$), percent deviation ($D$), Nash-Sutcliffe efficiency ($E$), and root mean square error (RMSE) for model performance evaluation, as in (Mahmood et al., 2016;García et al., 2008;Verma et al., 2010;Meenu et al., 2012). In addition, the simulated data were also compared with observed data graphically to investigate how well the model simulated the low and high flows in the basin. These indicators were calculated as below:

$$R^2 = \frac{\sum(Q_{obs} - \overline{Q_{obs}}) \times (Q_{sim} - \overline{Q_{sim}})}{\sqrt{\sum(Q_{obs} - \overline{Q_{obs}})^2 \times (Q_{sim} - \overline{Q_{sim}})^2}} \qquad 8$$

$$D\ (\%) = 100 \times \frac{Q_{sim} - Q_{obs}}{Q_{obs}} \qquad 9$$

$$E = 1 - \frac{\sum(Q_{sim} - Q_{obs})^2}{\sum(Q_{obs} - \overline{Q_{obs}})^2} \qquad 10$$

$$RMSE = \sqrt{\left(\frac{1}{n}\sum_{i=1}^{n}(Q_{sim} - Q_{obs})^2\right)} \qquad 11$$

where $Q_{obs}$ and $Q_{sim}$ describes observed and simulated streamflow, respectively. An $R^2$ value closer to 1 is the indication of good correlation between observed and modeled data. An $R^2$ value of 1 means model simulated data 100% same as observed. $D$ and $RMSE$ values should be closer to 0 for good results. Negative and positive values of $D$ show under and overestimation, respectively (Meenu et al., 2012). The values of $E$ stretches from 0 to 1. For good results, $E$ values should be positive and closes to 1, but a value which is negative and closer to 0 is not acceptable. $E$ values stretched between 0.36 and 0.75 are considered to be satisfactory and  greater than 0.75 are indication of good results (Van Liew and Garbrecht, 2003) (Meenu et al., 2012).

### 4.6 Impacts of climate and human on streamflow to Lake Chad

To quantify the impacts of climate variability and human activities on water resources in a basin, different kind of approaches have been reported in the literature. These are divided into four main categories, i.e., hydrological modelling approaches, as in (Hu et al., 2015), conceptual methods, as in (Mo et al., 2018), analytical approaches, as in (Tang and Lettenmaier, 2012), and experimental approaches, as in (Huang et al., 2003) and are reviewed comprehensively by (Dey and Mishra, 2017). In the present study, the hydrological simulation method was applied to separate and quantify the impacts of climate variability and human activities on rapidly decreasing inflow to Lake Chad. This method requires less information to process, e.g., we require only hydro-climatic data to simulate natural streamflow. However, this method requires long term time series to find out baseline period where human activities are considered to be minimum.

In this method, the first step is to explore the change point year in the streamflow time series using some statistical methods, and then the whole period is divided into a baseline period and an impacted period (mentioned above). Secondly, a hydrological model is calibrated and validated for the baseline period, and natural streamflow is simulated for the impacted period by forcing





only meteorological data to a hydrological model. Therefore, this can be referred as the hydrological responses to climate variability only (Hu et al., 2015). The changes between observed streamflows for a baseline period and for an impacted period describe the combined impacts of increased human activities and climate variability (Wang et al., 2013;Hu et al., 2015), and this can be calculated as below:

$$\Delta Q_T = \Delta Q_C + \Delta Q_H = Q_{OI} - Q_B \qquad 12$$

where $Q_B$ is the observed streamflow for a baseline period and $Q_{OI}$ for an impacted period. $\Delta Q_T$ demonstrates the total changes in streamflow, which is the sum of changes in streamflow due climate variability ($\Delta Q_C$) and changes in streamflow due to human activities ($\Delta Q_H$). In the previous studies, such as (Wang et al., 2013;Dey and Mishra, 2017;Hu et al., 2015;Wang et al.,

10 2008), $\Delta Q_C$ has been estimated by subtracting the average observed streamflow of baseline periods from average simulated streamflow of impacted periods. Since a model cannot simulate streamflow exactly the same as observed streamflow, and there are always some biases (e.g., model under or overestimates) between simulated and observed streamflows, which we can calculated during the model calibration and validation. For example, during the calibration and validation process in this study,

15 4–8% biases were obtained between simulated and observed streamflow (Table 5). This can mislead the original changes due to climate change and human activities. Therefore, in the present study, to reduce the effects of these biases, $\Delta Q_C$ was estimated by taking the difference between the simulated natural streamflow of the baseline period ($Q_{sB}$) and the natural streamflow of impacted period ($Q_{SI}$), as below:

$$\Delta Q_C = Q_{SI} - Q_{sB} \qquad 13$$

After estimation of $\Delta Q_C$, the change in streamflow due to human activities ($\Delta Q_H$) can be obtained by taking the difference of total changes and changes due to climate, as in (Dey and Mishra, 2017), as below:

$$\Delta Q_H = \Delta Q_T - \Delta Q_C \qquad 14$$

 Finally, the percentage contributions of climate variability ($P_C$) and human activities ($P_H$) to total streamflow changes are quantified, as in (Dey and Mishra, 2017;Wang et al., 2008), as below:

$$P_C(\%) = \frac{\Delta Q_C}{\Delta Q_T} \times 100 \ \ \text{and} \ \ P_H(\%) = \frac{\Delta Q_H}{\Delta Q_T} \times 100$$

In the present study, after the change point detection, HEC-HMS was calibrated and validated successfully for the baseline period (1951–1971), and then the natural streamflow was simulated by forcing the meteorological data for both the baseline period and the impacted period (1972–2013), at N'Djamena gauge in the CLRB. The simulated data from 1972–2015 was





divided into four decades: 1972–1981, 1982–1991, 1992–2001, and 2002–2011. Finally, the overall and decadal changes in streamflow due to climate variability, human activities, and due to both factors (climate and human) were assessed in the CLRB.

## 5 Results and discussion

### 5.1 Verification of CRU data for the study area

Before starting the main analysis, it is essential to evaluate CRU data with available good quality observed data. For this purpose, TM and PP of three climate stations (i.e., N'djamena, Sarh, and Moundou) were compared with CRU data using some statistical indicators (Table 2) and graphical plots (Fig. 3). The CRU data was extracted from the grids corresponding to the location of these stations. The statistics of monthly mean PP were calculated for the period of 1984–2015 for all three stations, and the statistics of monthly temperature were estimated for the periods of 1984–2014 at N'djamena, 1984–2001 at Moundou, and 1989–2013 at Sarh station. The coefficient of determination ($R^2$) between observed and CRU data were more than 0.9 for PP and above 0.95 in case of TM. *RMSE* ranged between 26 and 44 mm in case of PP, and 0.6 °C and 0.97 °C for TM. The mean and standard deviation from observed data were very close to CRU values for all stations (Table 2). The PP and TM of CRU underestimated a little on all stations except on Moundou in case of PP. Figure 3 shows that the mean monthly observed patters of both TM and PP were well followed by the CRU data at N'Djamena site. Thus, the results showed that CRU climatic variables can confidently be used for this study area.

### 5.2 Trend analysis for detection of climate change

Table 3 shows the Mann-Kendall test statistics ($Z$), Sen's slope ($Q$), significance level ($SL$), total change per 65 years ($Q_T$) of annual TM and PP for the period of 1951–2015 and inflow to the lake for the period 1951–2013 in the CLRB, the KYRB, the YENG, and the LFB. The positive and negative values of $Z$ show increasing (upward) and decreasing (downward) trends, respectively. *SL* value shows how statistically significant or strong the signals of trends are in the time series. *Q* value describes the magnitude or rate of change of trend per year in time series. The strength of trend signals was divided into four categories on the basis of significance level: (1) trends at α=0.1 (very weak signal of trend), (2) trends at α=0.05 (weak signal), (3) trends at α=0.01 (strong signal), and (4) trends at α=0.001 (very strong signal).

### 5.2.1 The Chari-Logone River basin

The whole CLRB basin was divided into 5 small basins according to their location by aggregating the sub-basins: the Logone River basin (LRB) located in the west of the CLRB, the Ouham River basin (ORB) in the south, the Chari River basin above Sarh (CRBAS) in the southeast, the Chari River basin below Sarh (CRBBS) in the northwest, and the Bahr Salamat basin (BSB) in the northeast, as shown in Fig. 2. PP was detected to decrease in all the sub-basins of the CLRB, with very weak to strong signals (Table 3). Two basins, the CRBAS and the BSB, showed strong signals of decreasing trends, with 11%, and





18% decrease for the period of 1951–2015, respectively, and two basins, the LRB and the CRBBS, showed weak decreasing trends, indicating 9% and 16% decrease in PP, respectively. However, the ORB showed very weak (α =0.1) decreasing trend, with 7% decrease in PP. On the whole, strong evidences of decreasing trends were estimated over the whole CLRB basin, showing 10% decrease in PP. Nkiaka et al. (2017) also showed decreasing trends in PP for the period of 1951–2000. However,

they conducted study only in the LRB, using observed point station PP. In contrast, in case of TM, very strong increasing trends were detected in all the sub-basins except in the ORB (weak signal). The highest increase in TM (1.9 °C) was estimated in the BSB (located in the north) for the whole period (1951–2015) and the lowest increase of 0.4 °C in the ORB (located in the south). In the whole basin, about 1.1 °C (0.16 °C decade$^{-1}$) of temperature was investigated to increase only in 65 years (1951–2015). Spatially, the south-western parts of the basin were observed to be less affected relative to the north-eastern parts

of the CLRB in case of increasing TM and decreasing PP (Table 3).

The plots of mean annual data along with Sen's Estimate lines (SE) are shown in Fig. 4a (for TM) and Fig. 5a (for PP), indicating definite increasing trends in TM and decreasing in PP in the CLRB. These plots also explore a clear picture of different rates of changes in the sub-basins. For example, a rapid rate of increase in TM was observed in the BSB and the CLRBAS (located in the north-eastern parts of the CLRB) relative to other sub-basins located in the southwest of the basin,

and the highest increasing rate in TM was detected in the BSB (Fig. 4a). It was also explored that, before the 1960s, TM in the BSB and the CRBAS was lower than the ORB (southern prat) but after that was higher. In case of precipitation, no abrupt changes were explored between the sub-basins, as in case of temperature. Instead of overall decrease in mean annual PP in the basin, there was sharp decrease in PP until the mid of the 1980s, but after that there was slight increase in PP up to 2015 (Fig. 5a). These plots also give some information on spatial variability of TM and PP across the basin. TM varies greatly from south

(e.g., low TM in the ORB) to north (e.g. high in the CRBBS) but a little from west (e.g. LRB) to east (CRBAS) (Fig. 4a) in the basin. In contrast, PP varies greatly from south (e.g. high PP in the ORB) to north (e.g. low in the BSB) and a little from west to east in the CLRB (Fig. 5a).

### 5.2.2 The Komadugu-Yobe River basin

The trend results of all sub-basins in the KYRB were combined into four main parts of the areas: the northern, the southern,

the eastern and the western parts. In the basin, similar kind of trend statistics were found in the annual TM and PP time series but with different magnitudes. In case of PP, weak decreasing signals were detected in the whole basin, with an overall decrease of 17%, higher than the CLRB (10%), for the period of 1951–2015. Strong decreasing trends in PP were explored in the northern parts of the basin and weak trends in the eastern and the western parts. Although the southern parts showed decrease in PP but not statistically significance. The highest decrease in PP (24%) was investigated in the northern parts. Similar to the

CLRB, very strong increasing trends were observed in TM in all parts of the basin, with an overall increase of 1.5 °C. The eastern parts, similar to the CLRB, showed the most increase in TM, about 1.6 °C (Table 3).

Increasing trends in TM and decreasing in PP are more clearly shown in Fig. 4b and Fig. 5b. These plots also show that the southern parts of the basin are colder than the northern parts, and PP declines from south to north in the basin.



### 5.2.3 The YENG basin

Like the KYRB, the trend results of all sub-basins in the YENG basin were combined into four major parts of the basin: the northern, the southern, the eastern, and the western parts. As in the CLRB and the KYRB, very strong increasing trends were observed in annual TM in all sub-basins, with 1.4 °C increase in the whole basin. Contrary to the CLRB, the KYRB, and the

LFB, the highest increase (1.6 °C) was explored in the western parts of the YENG basin. Although decreasing trends in annual PP were spotted in all regions of the basin, weak signals ($\alpha = 0.05$) were obtained in all parts of the basin except in the northern part (no significant trends). However, an overall decrease of 17%, similar to the KYRB and higher than the CLRB, was estimated in the basin for the whole period (Table 3). The graphical presentation of these trends shown in Fig. 4c and Fig. 4c gives more clear indication of increasing TM and decreasing PP in the basin. The increase rate of TM was faster in the western

parts than other parts of the YENG basin. Before 1990, the western parts were cooler than the northern parts but after that were warmer. On the other hand, the western parts were little wetter than the eastern parts in early decades, but later both parts received same amount of PP (Fig. 5c, SE lines western and eastern). Figure 5c also shows decrease in PP until the mid of 1980s, same like the KYRB and the CLRB, but increase after that in the whole basin.

### 5.2.4 The Lake Fitri basin

Since the LFB stretches more from east to west than north to south, the trends results of all sub-basins were categorized into 3 main parts: the eastern, middle and western parts (Table 3). In the LFB, there were strong signals of decreasing PP in the whole basin, with 18% decrease for 1951–2015. Similar to the CLRB, the KYRB, and the YENG, although there were decreasing trends in all regions of the basin, in the eastern parts, the trends were statistically non-significant, weak trends in the middle, and strong in the western region of the basin. Similar to other basins, very strong rising trends in TM were detected in all areas

of the basin, with an increase of 1.8 °C, which was the highest among other three main basins, i.e., the CLRB, the KYRB, and the YENG. Like other basin, the east-northern parts were changing faster than other parts in case of TM.  Figure 4d and Figure 5d show Sen's slops and annual TM and PP in the basin, respectively, which clearly indicates the presence of very strong increasing trend in TM and weak to strong decreasing signals in PP. Same like the KYRB, the CLRB, and the YENG, PP in the basin decreased until the mid of 1980s, but increased after that in all parts of the basin (Fig. 5d).

On the whole, very strong increasing trends in TM were explored in all four basins and weak to strong decreasing signals in case of PP. An average increasing TM of 1.45 °C (0.22 °C decade$^{-1}$) and decreasing PP of 16% (2.5% decade$^{-1}$) were explored in the study area, for the period of 1951–2015. Collins (2011) has shown increase in TM by 0.16 °C decade$^{-1}$ in Africa during 1979–2010. So the increasing rates of TM per decade are higher than the increasing rate to TM over the whole Africa. The similar results were explored by Funk et al. (2012) in Chad (country), they showed increase in TM by 0.8 °C (0.23 °C decade$^{-}$

$^{1}$) and decrease in precipitation by 13% (3.9% decade$^{-1}$) for 1975–2009 but only for rainy season (June–September), which are higher rates per decade than our results due to shorter period, rainy season and covering only some part of our study area. In all parts of basin, the maximum and the minimum rates of decrease in PP were explored to be 3.7% decade$^{-1}$ and 0.9% decade$^{-}$





[1], respectively, and the highest and lowest rates of increase in TM were 0.34 °C decade$^{-1}$ and 0.06 °C decade$^{-1}$. According to these rates, in the next 100 years, PP is expected to reduce by 9–37%, and TM is likely to increase by 0.6–3.4 °C in different parts of the study areas.

### 5.2.5 Inflow to Lake Chad

Temporal trends were also detected in annual inflow to LC from the CLR, the KYR, and the El-Beid River for the period of 1951–2013. Table 3 describes very strong (α=0.001) decreasing trends in inflow time series, showing alarming situation for LC. The flow from the CLR was perceived to decrease by 57% during the whole period, and by 74% and 69% decrease in flow from the KYR and the El-Beid River, respectively. Figure 6 shows a dramatic decrease in flow from all the rivers to the lake, though PP after the 1980s started increasing. Although decreasing signal in PP were not very strong and PP also started increasing after the mid of the 1980s, very strong signs of decreasing flow to the lake were observed, with an overall decrease of 67%. So we can conclude that decrease in flow to the lake is not entirely due to climatic variability (increase in temperature and decrease in precipitation). There are certainly some other factors that has caused decreasing flow to Lake Chad along with the climate variability in the basin. These factors, especially climate and human, are separated and quantified in the next sections.

### 5.3 Impacts of climate and human activities on streamflow to Lake Chad

#### 5.3.1 Change point analysis

Table 4 shows the change point years in hydro-climate time series in the Chari-Logone River basin. Both tests, the WLRT and SNHT, detected 1971 as a change point in PP and streamflow at α = 0.05. Although the SNHT showed 2005 and 1964 as change point in PP and streamflow, respectively, these were not statistically significant (α = 0.05). Statistical significant change points in TX, TN, and TM were detected in 2003, 1996, and 1993 by both tests. According to (Blench, 1997; Coe and Foley, 2001), most of the irrigation projects were started during the 1960s and 1970s. Figure 7 shows the test statistics calculated from both tests for streamflow at N'Djamena gauge. This shows a high peak of test statistics in the year 1971, providing the strong evidences of changes point.

#### 5.3.2 Calibration and validation

Table 5 shows the Nash efficiency ($E$), coefficient of determination ($R^2$), percent difference ($D$), and root mean square error ($RMSE$) calculated for performance evaluation of HEC-HMS for the calibration (1956–1965) and validation periods (1951–1955 and 1966–1971). During calibration, the values of $E$ and $R^2$ were 0.87, and $D$ was 3.85. During validation, $E$ and $R^2$ ranged between 0.84 and 0.88 for both validation periods, the $D$ values were less than 10%. The values of $RMSE$ were calculated to be less than 400 m$^3$ s$^{-1}$ during calibration and validation. Figure 8 shows the comparison of simulated flow by HEC-HMS and observed flow at N'Djamena gauge for calibration and validation periods. During calibration, the model well





overestimated during 1957 and 1964 and underestimated 1951,1960, and 1961 in peak months. Except 1956, the model well simulated the flow in low flow months. During the first validation period (1966–1971), model well overestimated in 1969 and a little in 1967 and 1970, and underestimated flow in 1966 and 1968 in peak flow months. During second validation period, model well overestimated in 1953 but underestimated in 1951, 1954 and 1955. During validation, the model also performed

better in low flow months than peak flow months. In the present study, we were more concerned about annual simulated flow rather than low or high flow. On the whole, the model performed reasonably well during calibration and validation according our objectives. According to (Van Liew and Garbrecht, 2003), $E$ values greater than 0.75 are greatly appreciated, and the values between 0.36 and 0.75 are referred to be satisfactory depending upon the objectives of the study.

**5.4 Hydro-climatic changes in the impacted period**

In the present study, we separated and quantified the decadal (i.e., 1972–1981, 1982–1991, 1992–2001, and 2002–2011) changes in streamflow to Lake Chad due to climate variability and human activities with respect to baseline period (1951–1971), and the decadal changes in climatic variables (i.e., TX, TN, TM, and PP) were also assessed relative to the baseline period.

**5.4.1 Climatic Changes in the impacted period**

Table 6 describes the decadal climatic changes in the CLRB with respect to 1951–1971. TM, TN, and TX were explored to increase in all decades, except 1972–1981, with respect to the baseline. The last decade was the hottest in all decades, where TM, TN, and TX were increased by 0.74 °C decade$^{-1}$, 1.16 °C decade$^{-1}$, and 0.9 °C decade$^{-1}$, respectively, with respect to the baseline period. It was also explored that rate of increase of TN was much higher than TX and TM, indicating rapid warming in the basin. In case of PP, negative changes were perceived in all decades in the CLRB, indicating decrease in PP with respect

to the baseline period. The highest decrease (45%) in PP was detected during 1982–1991, because worst drought was occurred in the 1980s, reported by different studies (Gao et al., 2011;Hansen and Przyborski, 2017;Zieba et al., 2017;Uche et al., 2015). After that, in the next two decades, PP started increasing relative to the second decade, as shown in Table 6. On the whole, the CLRB received 10% (2.5% decade$^{-1}$) less PP relative to the baseline, which was quite similar to the decrease in PP (15% 65yr$^{-1}$) calculated during trend analysis in case of decadal rate (2.3% decade$^{-1}$). Figure 9 also shows decrease in PP in the CLRB up

to the mid of 1980s and increase after that.

**5.4.2 Decreasing streamflow causes**

The changes in streamflow at N'Djamena hydrometric station due to both climate variability and human activities ($\Delta Q_T$), climate variability ($\Delta Q_C$), and human activities($\Delta Q_H$) are presented in Table 6. In all decades, negative changes in flow due to climate variability, human activities, and both (climate and human) were detected at N'Djamena gauge in the basin, indicating

clear reduction in inflow to Lake Chad. In the first decade (1972–1981), a 34% decrease in streamflow was estimated due to both climate variability and human activities, though decrease in PP was only 9%. It was estimated that 31% of total reduction



(34%) was due to climate variability and 69% due to human activities. We can say that 9% reduction in precipitation resulted in 31% reduction in streamflow to Lake Chad, though other factors, e.g., temperature, can also affect. In 1982–1991, total reduction in streamflow was 45%, of which 59% decline was due to climate variability and 41% due to human. Coe and Foley (2001) also showed total reduction in flow of 45% during almost the same period (1985–1994), of which 56% decline was due climate and 44% due to human, though we used different methodology. This was the only period that was mostly affected by climate variability because most devastating drought occurred during this period. In this period, precipitation was reduced by 18%, which was double of the previous decade and the highest decrease during the whole period (1972–2013) (Table 6), and also shown in Fig. 9.

In the third (1992–2001) and the fourth (2001–2011) decades, the total decline in streamflow was 36% and 45% with respect to baseline period, respectively. In the third decade, 39% decline of the total decline was due to climate variability and 61% due to human activities, and in the fourth decade, only 16% was due to climate and 84% due to human activities. After second decade decrease in streamflow due to climate variability stared improving in the next decades because PP started increasing (Fig. 9). On the whole, human activities mostly caused reduction in inflow to Lake Chad as compared to climate variability during 1972–2013, with 66% reduction in flow due to human activities and 34% due to climate (Table 6). Figure 9 shows annual PP, natural simulated flow, and observed streamflow at N'Djamena gauge. This shows that PP decreased in the CLRB until the 1980s and after that started increasing. The simulated natural flow well followed the observed streamflow until 1971 and after that the difference between natural flow and observed flow increased which was certain indication of extraordinary water extraction in the basin, especially for irrigation purposes. Although there was well increment in PP after the 1980s, there was a little or no increment in observed flow. Zhue et al. (2017) also showed a little or no increment in lake water level from 1990–2015 and concluded a stable water level since 1990s. According to the experiments of Gao et al. (2011), an increase of 106% of precipitation can recover the lake, without irrigation. They also showed that the lake can recover its size of 1963 with a continuous flow of 50 Km$^3$ yr$^{-1}$ for 10 years, about 40% higher than the average annual flow (29 km$^3$ yr$^{-1}$ for 1951–2013). So the lake can restore only by reducing the human activities, sustainable water resources management, and continuous increase in precipitation in the Chari-Logone River basin. From the above results and discussion, we recommend that it is better to transfer water from the Congo River to restore the Lake Chad quickly to its size of 1960s because it will not be easy to reduce the human activities in the future as the population has been increasing rapidly in the basin, and even with better human practices for irrigation, there still needs to transfer water from other river.

## 6 Conclusions

In the present study, hydro-climatic variability was examined by using trend analysis in the most active parts of the Lake Chad basin (i.e., Chari-Logone River basin (CLRB), the Komadugu-Yobe River basin (KYRB), the YENG (Yedseram, El-Beid, Ngadda, and Gubio Rivers) basin, and the Lake Fitri basin (LFB)), for the period of 1951–2015 (climatic variables) and 1951–2013 (inflow). Then the impacts of climate variability and human activities on decrease streamflow to Lake Chad were



separated and quantified in the Chari-Logone River basin, contributing more than 90% to the lake, using hydrological approach. In this study, we modified the equation used to calculate streamflow changes due to climate variability. So we calculated changes in streamflow due to climate variability by subtracting simulated streamflow of the baseline period from the simulated streamflow of the impacted period. In the previous studies, these changes were obtained by subtracting the

observed streamflow of the baseline period from the simulated streamflow of the impacted period, which can be influenced by biases between simulated and observed streamflow during calibration and validation. For hydrological approach, change point analysis were used to identify a baseline (stable or natural) period. So the streamflow data was divided into a baseline period (1951–1971) and an impacted period (1972–2013). HEC-HMS was calibrated for 1956–1965 and validated for 1951–1955 and 1966–1971 periods and was used to simulate flow for 1951–2013. Decadal changes (i.e., 1972–1981, 1982–1991, 1992–

2001, and 2002–2011) in streamflow due to climate and human activities were estimated with respect to the baseline period, at N'Djamena gauge on the Chari-Logone River. Since the monthly collected observed data in the basin was very scarce, monthly climatic data was extracted from the Climate Research Unit (CRU) and was converted in daily format using MODAWEC (Monthly to Daily Weather Convertor), as the requirements of HEC-HMS.

Trend analysis showed very strong signals (α=0.001) of increasing trend in temperature in all basins. In contrast, decreasing

signals were detected in precipitation, however, signals ranged between very weak (α=0.1) to strong (α=0.01). An overall increase of 1.1 °C, 1.5 °C, 1.4 °C, and 1.8 °C was observed in the CLRB, KYRB, YENG, and LFB, respectively, during 1951–2015. On the other hand, precipitation was decreased by 10%, 17%, 17%, and 18% in the CLRB, KYRB, YENG, and LFB, respectively. The highest rise in temperature (2.2 °C) was explored in the eastern parts of the LFB and the highest decline in precipitation (24%) in the northern parts of the KYRB, for the whole period 1951–2015. The north-eastern parts of the study

areas were affected more than other parts in case of increasing temperature and decreasing precipitation. Using the rates of decrease in precipitation and increase in temperature, we can say that in the next 100 years, precipitation is likely to reduce by 9–37%, and temperature is anticipated to increase by 0.6–3.4 °C in different parts of the study areas. On the other hand, very strong decreasing trends were observed in streamflow to Lake Chad from the Chari-Logone, the Komadugu-Yobe, and the El-Beid Rivers, with a decline of 57%, 74%, and 69%, respectively, for the period of 1951–2013.

In case of decadal changes in the impacted period, the last decade (2002–2011) was the hottest decade relative to 1951–1971 in the CLRB, where mean, maximum, and minimum temperatures were increased by 0.9 °C, 0.74 °C and 1.16 °C, an alarming situation, and the second decade (1982–1991) was the driest, with 18% decrease in precipitation. On the whole, temperature was increased by 0.31 °C and precipitation decreased by 10% in 1972–2013 relative to 1951–1971.

Decreasing streamflow to the lake due to both climate and human activities were ranged between 34% and 45% in different

decades. The most reduction in streamflow was estimated due to human activities in all decades, ranging between 61% and 84%, except in 1982–1991, where 59% decline was explored due to climate variability because this period was affected mostly by a long devastating drought. After the second decade, the impacts of climate variability on streamflow were reduced because of increasing precipitation. On the whole, an average decrease of 40% was estimated due to climate variability and human activities for the period of 1972–2013, of which 66% of total decline was due to human activities and 34% due to climate



variability. Thus, human activities such as irrigation withdrawal in the Lake Chad basin caused most reduction in streamflow to the lake than climate variability. So inflow to the lake can be improved by reducing the human activities and sustainable water resources management. However, it looks not possible to recover the lake's size of 1960s (25,000 km$^2$) in coming recent years by just considering the better water management practices because population has been increasing rapidly in the basin.

So, we recommend that, it is better to transfer water from the Congo River for quick recovery of lake Chad. In this study, we quantified only impacts of human activities as a whole, it is recommended to assess impacts of different human activities, e.g., irrigation withdrawal, dam construction, domestic water use, industrial water use etc., separately, for much better understanding about which human intervention affecting most the inflow to the Lake Chad.

**Author contribution:** The first author, Dr Rashid Mahmood, designed and conducted this study under the supervisor of the second author, Professor JIA Shaofeng.

**Competing interests:** The authors declare that they have no conflict of interest.

**Acknowledgments:** This study is the part of water transfer project in the LCB. This research was funded by the National Key Research and Development Program of China (2016YFC0401307) and Power China International Limited. Thanks are given 15 to the National Aeronautics and Space Administration (NASA), the Climatic Research Unit (CRU), and the Lake Chad Basin Commission (LCBC) for providing their valuable datasets.

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



**Table 1.** Basic characteristics of the study area, mean values for the period of 1951–2013 (inflow) and 1951–2015 (temperature and precipitation).

| | Area (km²) | % area of LCB | % Inflow to Lake Chad | Annual inflow flow to Lake Chad (m³·s) | PP (mm) | TM (°C) |
|---|---|---|---|---|---|---|
| | | | mean (min–max) | mean (min–max) | | |
| Lake Chad basin | $2.5 \times 10^2$ | — | — | — | 465 | 26.7 |
| Conventional basin | 967 000 | 40 | — | — | 677 | 27.5 |
| Chari-Logone basin | 620 000 | 24.8 | 91.5 (86–98) | 931 (586–1356) | 935 | 26.7 |
| Komadugu-Yobe basin | 145 833 | 5.8 | 4.1 (0.3–6) | 74 (4–136) | 583 | 27.1 |
| YENG basin | 81 334 | 3.3 | 1.1 (0.4–1.6) | 20 (4–41) | 663 | 27.9 |
| Lake Fitri basin | 64 035 | 2.6 | N/A | N/A | 527 | 28.3 |

*LCB* Lake Chad basin; *YENG* Yedzeram, El-Beid, Ngadda, and Gubio*; TM* mean temperature*; PP* precipitation





**Table 2.** Comparison of the monthly observed and CRU temperature and precipitation for the period of 1984–2013, in the Lake Chad basin.

| | Precipitation (mm) | | | Mean temperature (°C) | | |
|---|---|---|---|---|---|---|
| | Moundou | N'Djamena | Sarh | Moundou | N'Djamena | Sarh |
| $R^2$ | 0.90 | 0.94 | 0.91 | 0.96 | 0.97 | 0.90 |
| RMSE | 44.3 | 26.0 | 42.9 | 0.65 | 0.81 | 0.97 |
| Observed STD | 103.6 | 73.1 | 102.4 | 2.31 | 3.36 | 2.18 |
| CRU STD | 106.1 | 65.1 | 91.3 | 2.26 | 3.25 | 2.14 |
| Observed mean | 86.0 | 47.3 | 80.8 | 27.47 | 29.05 | 28.20 |
| CRU mean | 87.3 | 42.5 | 77.2 | 26.86 | 28.68 | 27.92 |

STD *standard deviation*, CRU *climate research unit*, RMSE *root mean square error*





**Table 3.** Mann-Kendall test statistic (Z), Sen's slope (Q), total change ($Q_T$), and significance level (SL) of trends in the Chari-Logone, the Komadugu-Yobe, the YENG, and the Lake Fitri basins obtained from precipitation, temperature and inflow to Lake Chad.

| Basin and sub-basins | Temperature | | | | Precipitation | | | |
|---|---|---|---|---|---|---|---|---|
| | Z | SL | Q (°C yr$^{-1}$) | $Q_T$ (°C 65yr$^{-1}$) | Z | SL | Q (mm yr$^{-1}$) | $Q_T$ (% change 65yr$^{-1}$) |
| **Chari-Logone River basin** | | | | | | | | |
| Logone basin, west | 4.5 | *** | 0.01 | 0.70 | -2 | * | -1.626 | -9 |
| Chari above Sarh, East-south | 6.06 | *** | 0.017 | 1.10 | -2.97 | ** | -1.891 | -11 |
| Bahr Salamat basin, east-north | 7.39 | *** | 0.029 | 1.90 | -2.76 | ** | -1.894 | -18 |
| Ouham Basin, south | 2.53 | * | 0.006 | 0.40 | -1.85 | + | -1.442 | -7 |
| Chari below Sarh, north-west | 6.3 | *** | 0.018 | 1.20 | -2.45 | * | -1.878 | -16 |
| **Average** | **6.02** | **\*\*\*** | **0.017** | **1.10** | **-2.73** | **\*\*** | **-1.621** | **-10** |
| **Komadugu-Yobe River basin** | | | | | | | | |
| Northern part | 6.5 | *** | 0.021 | 1.40 | -2.73 | ** | -1.819 | -24 |
| Southern part | 6.38 | *** | 0.022 | 1.40 | -1.38 | | -0.962 | -6 |
| Eastern part | 6.45 | *** | 0.024 | 1.60 | -2.58 | * | -2.101 | -22 |
| Western part | 6.17 | *** | 0.022 | 1.40 | -2.58 | * | -1.901 | -17 |
| **Average** | **6.45** | **\*\*\*** | **0.023** | **1.50** | **-2.53** | **\*** | **-1.922** | **-17** |
| **YENG River basin** | | | | | | | | |
| Northern part | 6.46 | *** | 0.024 | 1.50 | -1.24 | | -0.831 | -12 |
| Southern part | 6.31 | *** | 0.019 | 1.30 | -2.46 | * | -1.549 | -13 |
| Eastern part | 6.46 | *** | 0.022 | 1.40 | -2.35 | * | -1.432 | -15 |
| Western part | 6.54 | *** | 0.024 | 1.60 | -2.45 | * | -2.265 | -22 |
| **Average** | **6.3** | **\*\*\*** | **0.021** | **1.40** | **-2.13** | **\*** | **-1.644** | **-17** |
| **Lake Fitri basin** | | | | | | | | |
| Eastern part | 6.66 | *** | 0.034 | 2.20 | -1.35 | | -0.964 | -12 |
| Western part | 6.48 | *** | 0.028 | 1.40 | -3.14 | ** | -1.868 | -18 |
| Middle part | 5.89 | *** | 0.021 | 1.80 | -2.52 | * | -1.682 | -18 |
| **Average** | **6.41** | **\*\*\*** | **0.028** | **1.80** | **-2.81** | **\*\*** | **-1.651** | **-18** |
| *Average Changes of all basins* | *6.30* | | *0.022* | *1.45* | *-2.55* | | *-1.710* | *-16* |

| | Inflow to Lake Chad | | | |
|---|---|---|---|---|
| | Z | SL | Q (m$^3$ s$^{-1}$) | QT (% change 63yr$^{-1}$) |
| Chari-Logone River | -6.83 | *** | -11.23 | -57 |
| El-Beid River | -5.03 | *** | -1.36 | -74 |
| Komadugu-Yobe River | -4.92 | *** | -0.32 | -69 |
| **Average Changes** | **-5.59** | | **-4.30** | **-67** |

*+ very weak signal of trend (α=0.1), * weak signal of trend(α=0.05), ** strong signal of trend (α=0.01), *** very strong signal of trend (α=0.001)*





**Table 4.** Change points detected in hydro-climatic time series in the Chari-Logone River basin ($\alpha = 0.05$).

| Time series | WLRT test | | SNHT test | |
|---|---|---|---|---|
| | Change point | Test Statistics | Change Point | Test Statistics |
| Precipitation | 1971 | 5.04 | 1971 | 22.436 |
| | | | 2005 | 8.813 |
| Discharge | 1971 | 12.252 | 1971 | 80.774 |
| | | | 1964 | 9.898 |
| Max Temperature | 2003 | 6.546 | 2003 | 34.427 |
| Min temperature | 1996 | 9.308 | 1996 | 57.652 |
| Mean Temperature | 1993 | 6.796 | 1993 | 36.505 |
| | | | 2004 | 7.715 |

SNHT *Standard Normal Homogeneity Test,* WLRT *Worsley Likelihood Ration Test*





**Table 5.** Performance indicators for calibration and validation at N'Djamena gauge on Chari-Logone River.

|  | Calibration | Validation |  |
|---|---|---|---|
|  | 1956–1965 | 1966–1971 | 1951–1955 |
| Nash Efficiency (E ) | 0.87 | 0.84 | 0.88 |
| Coefficient of Determination (R²) | 0.87 | 0.85 | 0.88 |
| Percent difference (D) % | 3.85 | 7.34 | −0.88 |
| RMSE (m³/s) | 381.33 | 385.26 | 394.30 |





**Table 6.** Changes in temperature (°C) and precipitation (%), and changes in flow (%) due to climate variability, human activities, and both (climate and human activities) with respect to 1951–1971 in the Chari-Logone River basin.

| Decade | ΔTX (°C) | ΔTN (°C) | ΔTM (°C) | ΔP (%) | ΔQ$_T$ (%) | ΔQ$_C$ (%) | ΔQ$_H$ (%) | P$_C$ (%) | P$_H$ (%) |
|---|---|---|---|---|---|---|---|---|---|
| 1972–1981 | -0.29 | -0.01 | -0.19 | -9 | -34 | -11 | -23 | 31 | 69 |
| 1982–1991 | 0.02 | 0.35 | 0.14 | -18 | -45 | -27 | -18 | 59 | 41 |
| 1992–2001 | 0.15 | 0.56 | 0.31 | -8 | -36 | -14 | -22 | 39 | 61 |
| 2002–2011 | 0.74 | 1.16 | 0.9 | -9 | -45 | -7 | -38 | 16 | 84 |
| **1972–2013** | 0.17 | 0.55 | 0.31 | -10 | -40 | -14 | -26 | 34 | 66 |

ΔTX *changes in maximum temperature,* ΔTN *changes in minimum temperature,* ΔTM *changes in mean temperature,* ΔP *changes*

*in precipitation,* ΔQ$_T$ *total changes in streamflow due to climate variability and human activities,* ΔQ$_C$ *total changes in streamflow*

*due to climate variability,* ΔQ$_H$ *total changes in streamflow due to human activities,* $P_C = 100 \times (\Delta Q_C / \Delta Q_T)$, $P_H = 100 \times (\Delta Q_H / \Delta Q_T)$





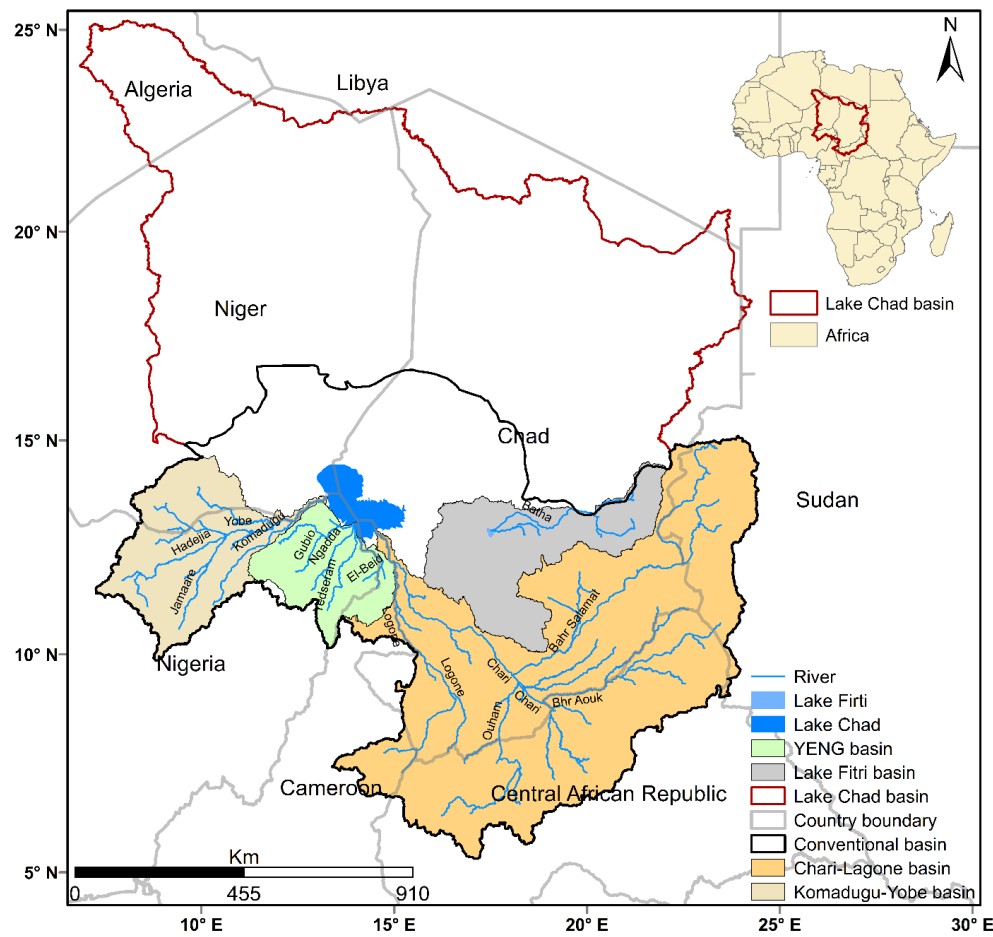

**Figure 1.** Location map of the study area (i.e., the Chari-Logone, the Komadugu-Yobe, the YENG, and the Lake Fitri basins) in the Lake Chad basin. Where YENG stands for the Yedseram, El-Beid, Ngadda, and Gubio River basins.




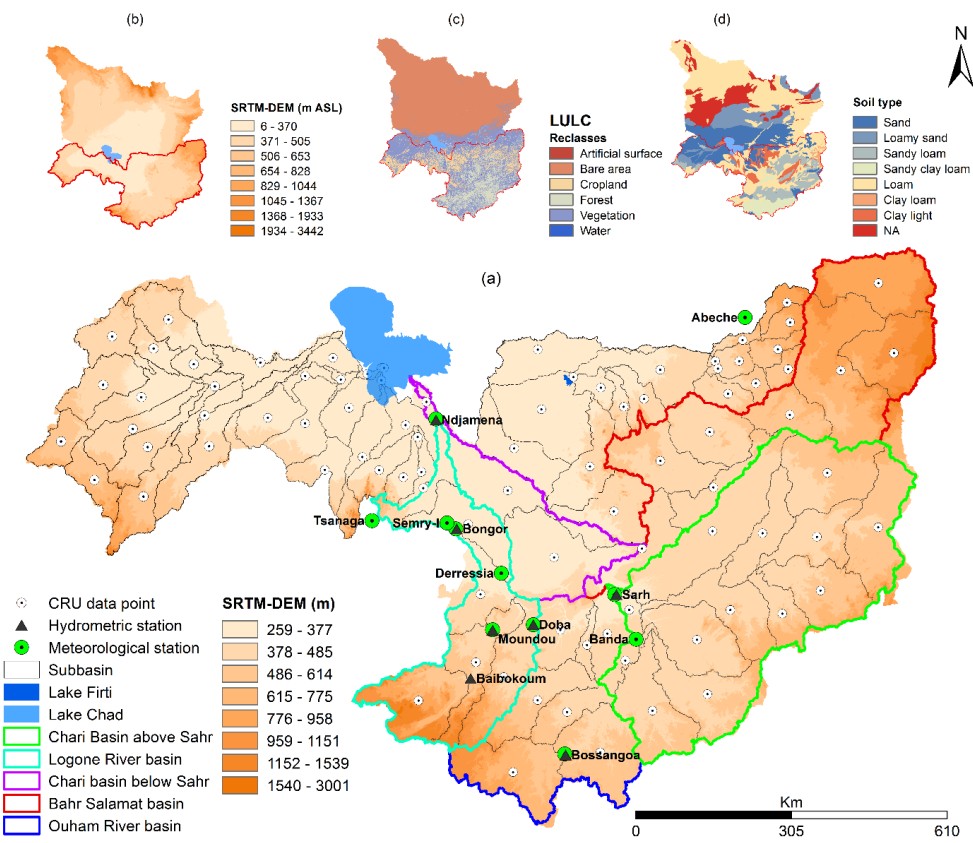

**Figure 2.** Description of land covers, soil types, elevation, CRU data, available hydro-meteorological stations, and sub-basins in the study area. Overlapping hydrometric and climatic stations have same names.





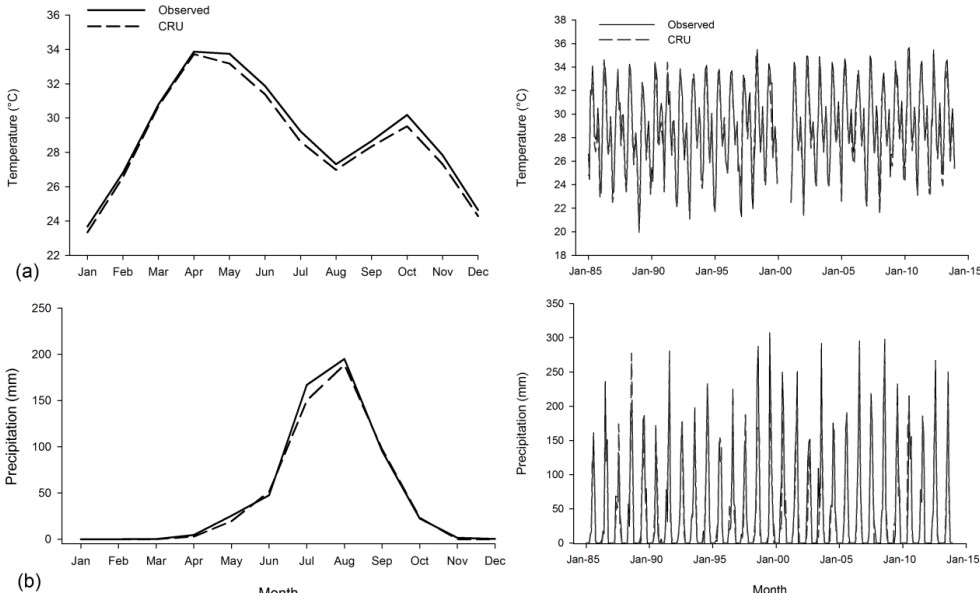

**Figure 3.** Comparison of CRU against the observed (a) temperature and (b) precipitation at N'djamena site for the period of 1984–2001.




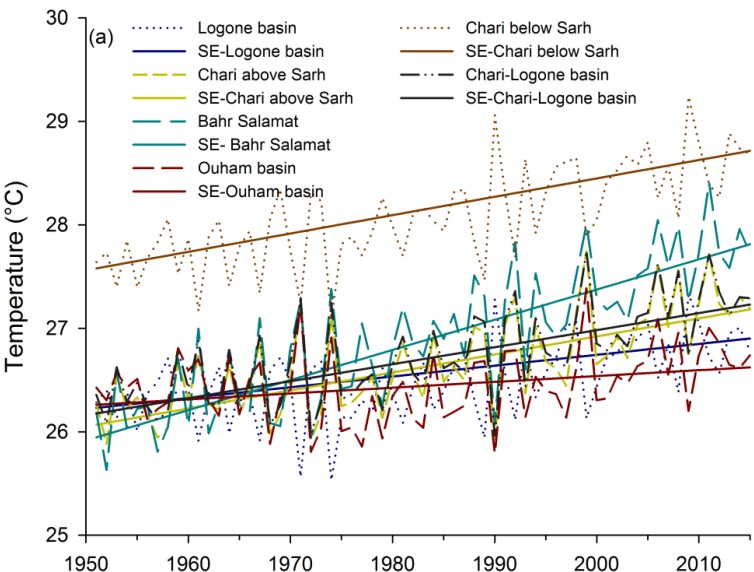

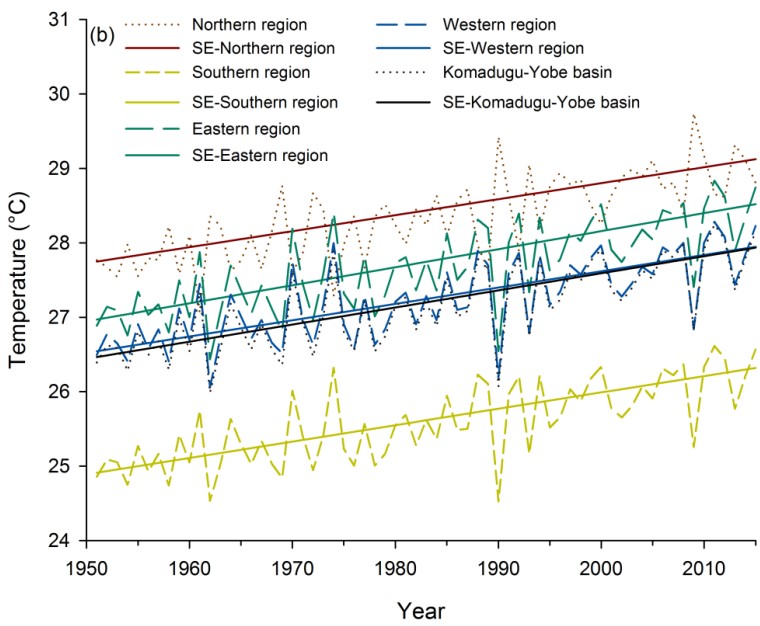

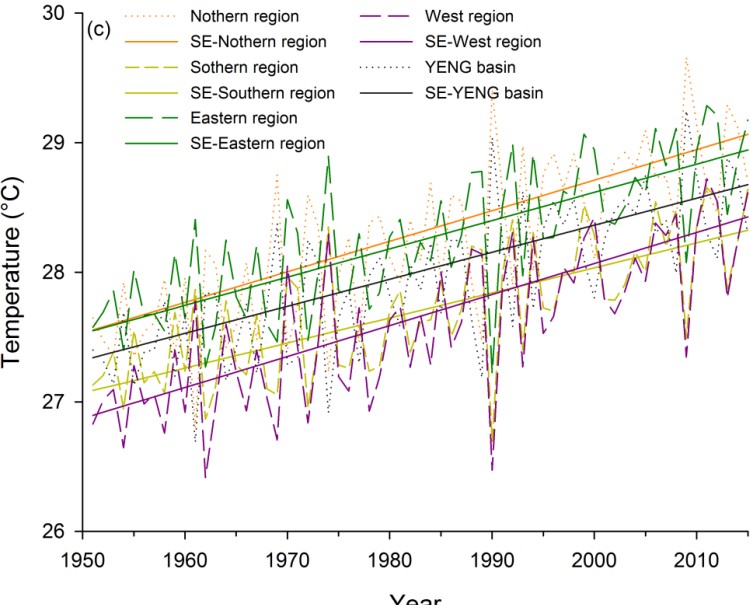

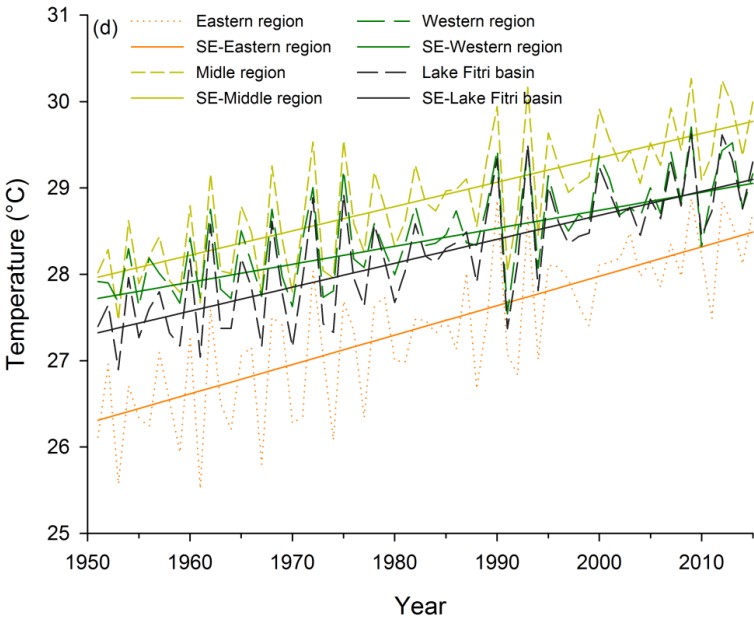

**Figure 4.** Annual temperature along with trend lines (Sen's slope estimates (SE)) in the (a) sub-basins of the Chari-Logone basin, (b) sub-basins of the Komadugu-Yobe basin, (c) sub-basin of the YENG basin and, and (d) sub-basin of the Lake Fitri basin.





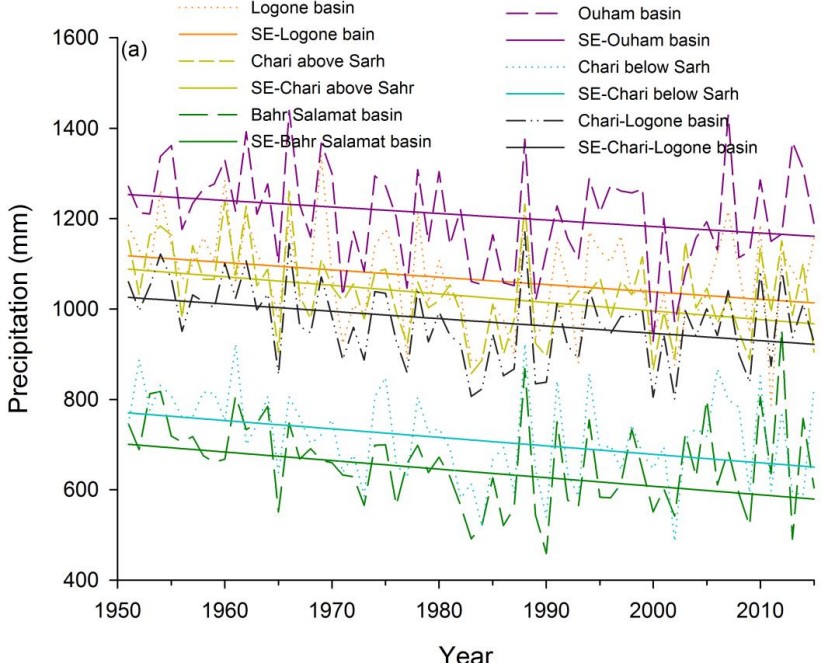

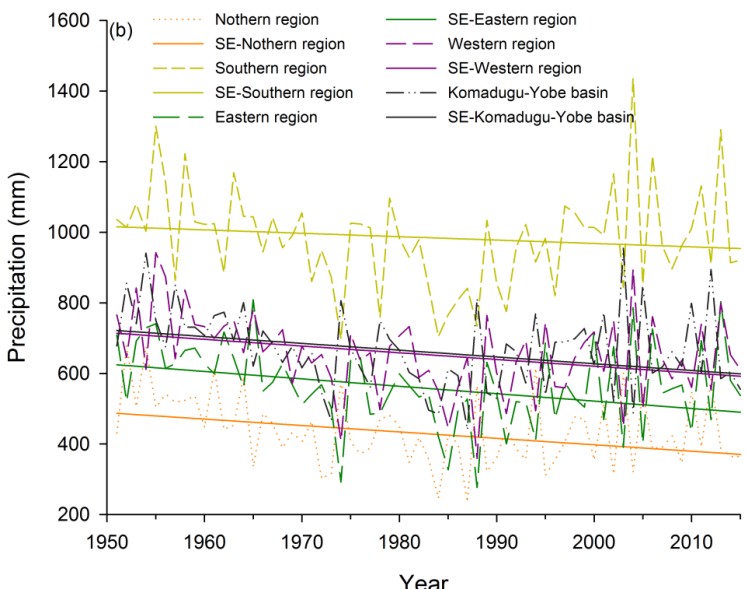




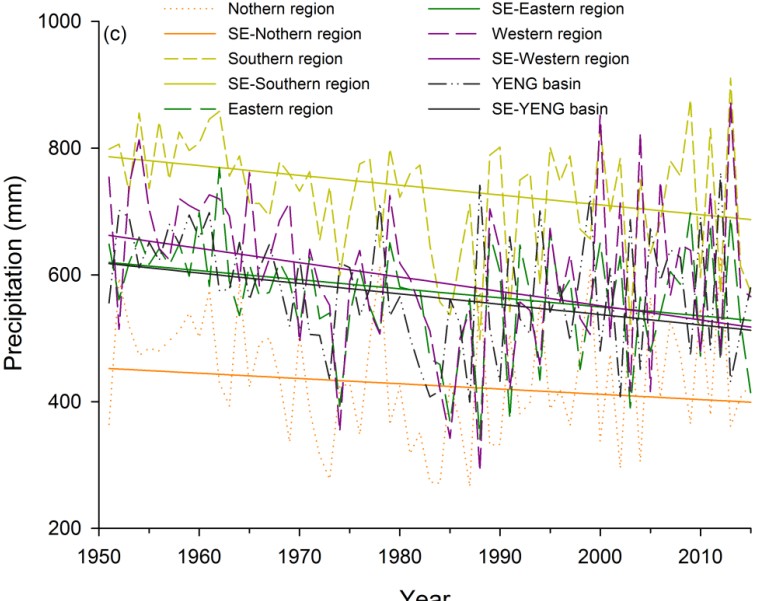

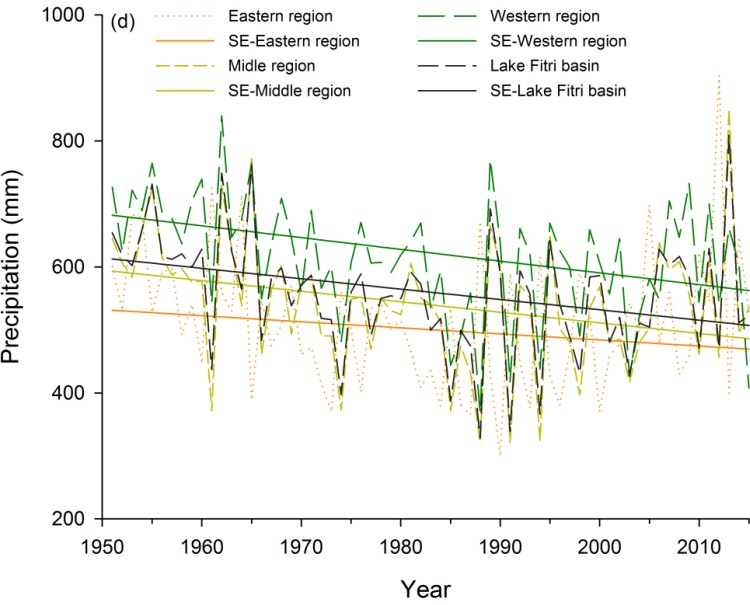

**Figure 5.** Annual precipitation (mm) along with trend lines (Sen's slope estimates (SE)) in the (a) sub-basins of the Chari basin, (b) sub-basins of the Komadugu-Yobe basin, (c) sub-basin of the YENG basin and, and (d) sub-basin of the Lake Fitri basin.





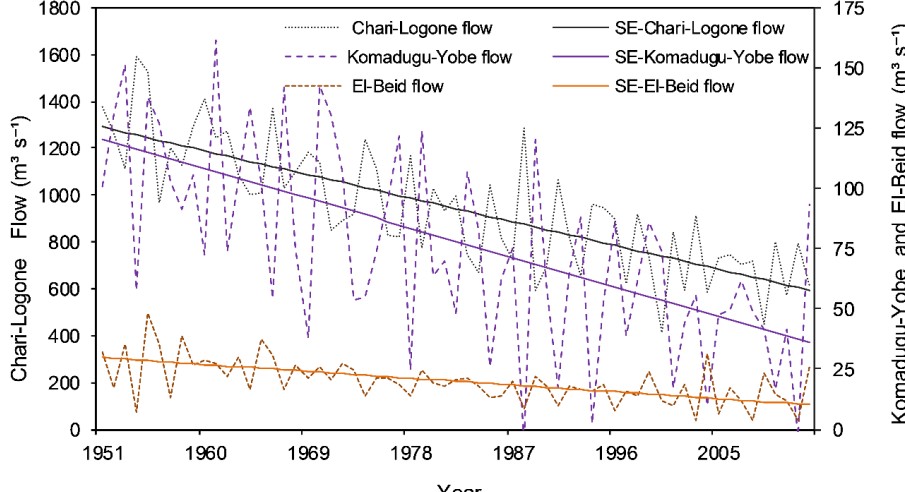

**Figure 6.** Inflow (m³s⁻¹) to Lake Chad from the Chari River, the El-Beid River, and the Komadugu-Yobe River along with the trend lines (Sen's slope estimates (SE)).





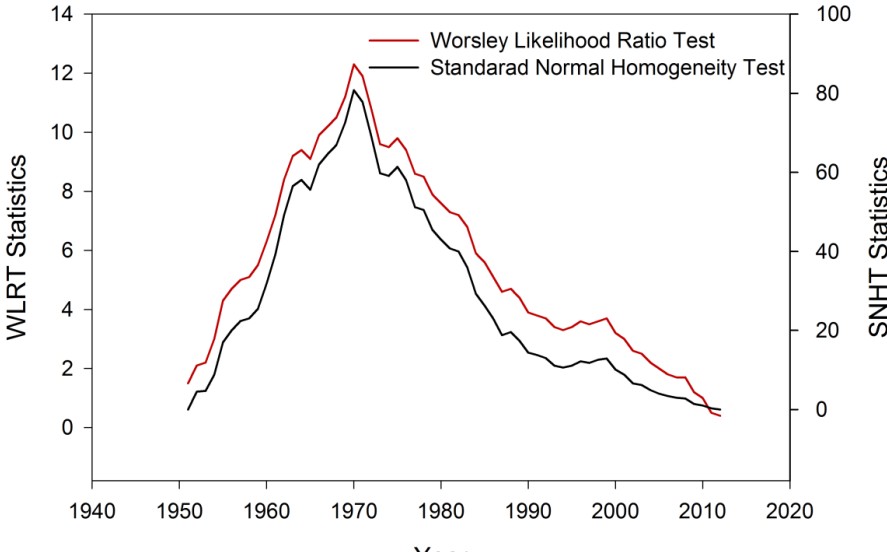

**Figure 7.** Change point year in streamflow at N'djamena gauge located on the Chari-Logone River detected by Worsley Likelihood Ratio Test (red line) and Standard Normal Homogeneity Test (black line).




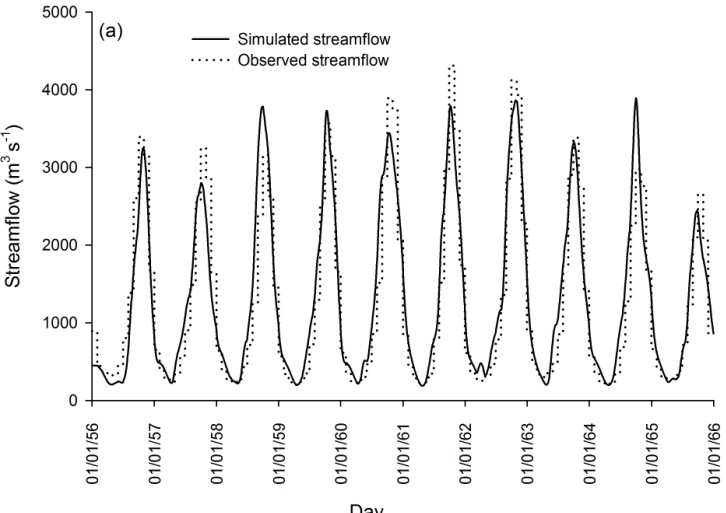

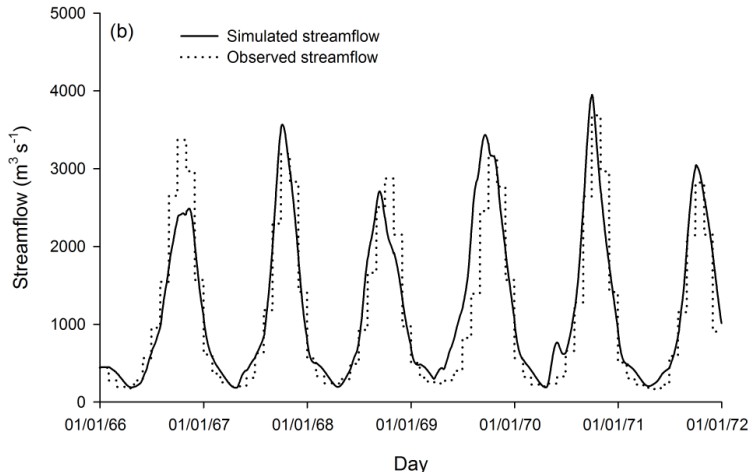





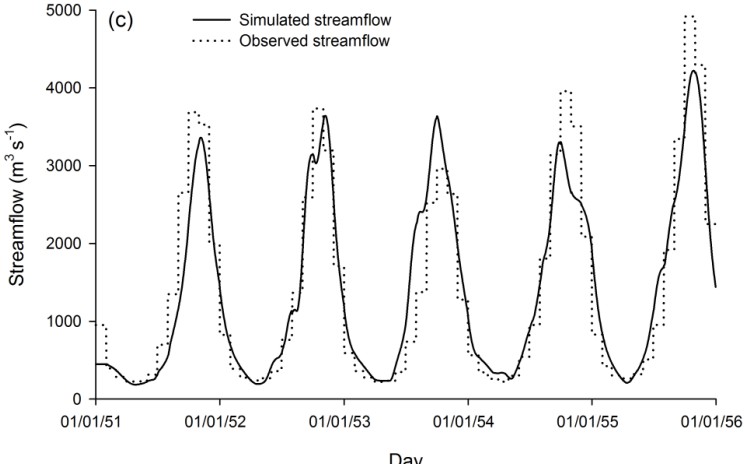

**Figure 8.** Comparison of simulated and observed streamflow at N'djamena gauge for the (a) calibration period (1955–1965) and (b and c) validation periods (1966–1971 and 1951–1955) in the Chari-Logone River basin



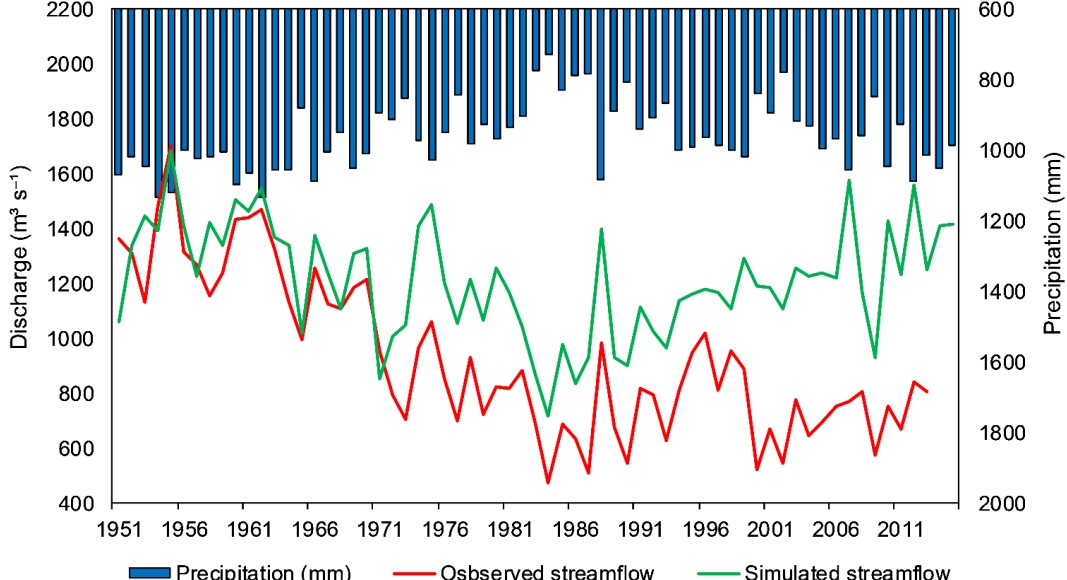

**Figure 9.** Analyzing annual precipitation of the basin along with naturalized (simulated) and observed flows at N'Djamena station in the Chari-Logone River basin.

