# Peer review of "Analysis of causes of decreasing inflow to the Lake Chad due to climate variability and human activities"

_Hydrology and Earth System Sciences, 2018_

## Referee Comment (RC1) · Anonymous Referee #1 · 17 Apr 2018

In order to understand the recent changes in Lake Chad, the authors have gathered the available climatic and hydrological data ( observed and from international databases) from 1951 to 2015. They identified a change point in the sequence of rainfall and Chari River discharge (around 1971). The trend before 1971 is attributed to climatic changes, while the trend after 1971 is attributed to both climatic and human impacts. According to the authors, the human impact ( mostly from irrigation development around 1970 in Nigeria) represents a major part of the decrease in discharge during the second period.

The attribution of the main change in Chari discharge mainly to irrigation development is not substantiated and seems debatable for several reasonsÂă: i) the large nigerian

irrigation schemes developed in the early 1970s have never been in operation because of the rapid recession of Lake Chad at that time and ii) the amount of irrigation impact involved by the authors ( in the order of 10 km3/year) would feed some 500 000 ha of irrigated land that are not identified on the ground nor by satellite observations. Although the rain decrease is clearly described, its impact on vegetation and land cover, on ground water level or soil surface should probably be discussed.

In a region where quite a number of frzncophone auhtors have published papers, onle one out of about 60 references quote a paper in french.

See also comments in the manuscript

Please also note the supplement to this comment:
https://www.hydrol-earth-syst-sci-discuss.net/hess-2018-139/hess-2018-139-RC1-supplement.pdf

**Supplement:**

[Figure]

**Analysis of causes of decreasing inflow to the Lake Chad due to climate variability and human activities**

Rashid Mahmood, Shaofeng JIA

Key Laboratory of Water Cycle and Related Land Surface Processes/Institute of Geographic Science and Natural Resources
5   Research, Chinese Academy of Sciences, Beijing 100101, China

*Correspondence to*: Shaofeng JIA, Rashid Mahmood, (jiasf@igsnrr.ac.cn, rashi1254@gmail.com, )

**Abstract.** In the 1960s, Lake Chad was the world's sixth largest water body, which has since shrunk dramatically from a surface area of 25,000 km$^2$ to only 2,000 km$^2$ in the following 40 years. In the present study, hydro-climatic variability in the Chari-Logone, Komadugu-Yobe, YENG (Yedseram, El-Beid, Ngadda and Gubio basins) as well as Lake Fitri basins and

10   decreasing streamflow to Lake Chad due to climate variability and human activities were separated and quantified using trend analysis, change point analysis, and hydrological approach, for the period of 1951–2015. The results showed very strong signals (α=0.001) of increasing trend in mean temperature, with an average increase of 1.4 °C, and very weak (α=0.1) to strong (α=0.01) decreasing signals in precipitation, with an average decrease of 15%. In case of streamflow to Lake Chad, very strong decreasing trends were observed, showing 67% reduction for the whole period. The north-eastern parts were most affected

15   parts in case of increasing temperature and decreasing precipitation. Decreasing flow due to both climate variability and human activities  ranged from 34% to 45% in different decades, from 1972 to 2013. On the whole, a 66%  total decline in streamflow was observed due to human activities and 34% due to climate variability. Most reduction in streamflow (59%) due to climate variability was explored only during 1982–1991 because a devastating drought  occurred during this period. Since human activities caused most reduction in streamflow to Lake Chad than climate, inflow to the lake can be improved by

20   reducing or properly managing the human activities and using sustainable water resources management.

**1 Introduction**

Lake Chad (LC), one of the biggest lakes in the world, is located in the Lake Chad basin (LCB), the largest endorheic basin in Africa (Gao et al., 2011). LC straddles the borders of Chad, Nigeria, Niger, and Cameroon, as shown in Fig. 1. As of 2011, it provides livelihoods to more than 30 × 10⁶ people across the basin. It is a vital source of freshwater and fishing and also

25   provides water for pastoral and agricultural land. In the 1960s, LC was the world's sixth largest water body, with a surface area of 25,000 km². However, in the subsequent 40 years, it has dramatically reduced to 2,000 km² (LCBC, 2011;Coe and Foley, 2001) and even about 300 km² in the 1980s (Gao et al., 2011). In 1975, the lake was divided into two parts (i.e., the northern pool and the southern pool) because of devastating drought over the African Sahel belt. Since then, the northern pool has been rarely and partly inundated (Lemoalle et al., 2012). According to Zhu et al. (Zhu et al., 2017), the southern pool of

**Comment [mm1]:** Can lake Chad ( population about 2 M people) provide food to almost the whole population of its basin ( 30 M people) ? Please quote and discuss the source

**Comment [mm2]:** It seems that there is a debate on this figure, whether the open water area or the total area with the inundated marshes is considered.

[revised manuscript text omitted]

$$() = _1 + ( \frac{^2}{} )_2 \qquad ^2 \qquad = 1,2, \dots, \qquad\qquad 6$$

where

$$_1 = \frac{1}{} \sum_{=1} {}^{-\bar{}} \qquad\qquad _2 = \frac{1}{-} \sum_{=+1} {}^{-\bar{}}$$

$$= \max_{1 \le \le} ()$$

Test statistic *T(k)* compares the mean of the first *k* values with the mean of the last *n-k* values. The highest value () obtained from *T(k)* statistics is the indication of the change point in the time series. To check the change point statistically significance, is compared with the critical values (given in Table III of (Wijngaard et al., 2003)) at specified significance
10   level (e.g., α=0.05). and⁻ are standard deviations and mean values of time series.

**Comment [mm15]:** English not clear

**4.3.2 Worsley Likelihood Ration test**

Worsley Likelihood Ration test (WLRT) developed by (Worsley, (1979) explores the most likely position of a change in mean value of a time series ($y_1$, $y_2$,…,$y_n$). In this method, deviations from the mean are summed up at each point (k=1, 2,…,n), and then weights are given to each sum according to their positions in the time series. This method is used frequently to explore

**Comment [mm16]:** English not clear

15   the position of jumps in time series, as in (Buishand, 1982;Vezzoli et al., 2012;Dollar et al., 2006).The formulation of this test explained by (Buishand, 1982) is given below:

$$=^* \frac{}{(( - ))^{0.5}} \qquad\qquad 7$$

where

$$^* = \sum_{=1} ( {}^{-\bar{}} )$$

$$= \max_{1 \le \le} |{}^{**}|$$

$$= \frac{( - 2)^{0.5}}{(1 - {}^2)^{0.5}}$$

[revised manuscript text omitted]

Comment [mm21]: Already stated

Comment [mm22]: Thyssen is maybe not the best approach in regions where there is a strong North-South gradient

Comment [mm23]: Not clear

Comment [mm24]: You here decide that the change point , which may be considered as a change in the climate ( rainfall) is also a point of change in the impact
This is an essential statement
Should be discussed more thoroughly

In the present study, the simulated flow was compared with observed data using four commonly used statistical indicators, i.e., coefficient of determination ($R^2$), percent deviation ($D$), Nash-Sutcliffe efficiency ($E$), and root mean square error (RMSE) for model performance evaluation, as in (Mahmood et al., 2016;García et al., 2008;Verma et al., 2010;Meenu et al., 2012). In addition, the simulated data were also compared with observed data graphically to investigate how well the model simulated

5 the low and high flows in the basin. These indicators were calculated as below:

$$R^2 = \frac{\sum( -\bar{}) \times ( -\bar{})}{\sqrt{\sum( -\bar{})^2 \times ( -\bar{})^2}} \qquad 8$$

$$D(\%) = 100 \times \frac{ - }{} \qquad 9$$

$$E = 1 - \frac{\sum( - )^2}{\sum( -\bar{})^2} \qquad 10$$

$$RMSE = \sqrt{\left(\frac{1}{}\sum_{=1}( - )^2\right)} \qquad 11$$

10 where $Q_{obs}$ and $Q_{sim}$ describes observed and simulated streamflow, respectively. An $R^2$ value closer to 1 is the indication of good correlation between observed and modeled data. An $R^2$ value of 1 means model simulated data 100% same as observed. $D$ and $RMSE$ values should be closer to 0 for good results. Negative and positive values of $D$ show under and overestimation, respectively (Meenu et al., 2012). The values of $E$ stretches from 0 to 1. For good results, $E$ values should be positive and closes to 1, but a value which is negative and closer to 0 is not acceptable. $E$ values stretched between 0.36 and 0.75 are

15 considered to be satisfactory and when greater than 0.75 are indication of good results (Van Liew and Garbrecht, 2003) (Meenu et al., 2012).

**4.6 Impacts of climate and human on streamflow to Lake Chad**

To quantify the impacts of climate variability and human activities on water resources in a basin, different kind of approaches have been reported in the literature. These are divided into four main categories, i.e., hydrological modelling approaches, as

20 in (Hu et al., 2015), conceptual methods, as in (Mo et al., 2018), analytical approaches, as in (Tang and Lettenmaier, (2012), and experimental approaches, as in (Huang et al., 2003) and are reviewed comprehensively by (Dey and Mishra, 2017). In the present study, the hydrological simulation method was applied to separate and quantify the impacts of climate variability and human activities on rapidly decreasing inflow to Lake Chad. This method requires less information to process, e.g., we require only hydro-climatic data to simulate natural streamflow. However, this method requires long term time series to find out

25 baseline period where human activities are considered to be minimum.
In this method, the first step is to explore the change point year in the streamflow time series using some statistical methods, and then the whole period is divided into a baseline period and an impacted period (mentioned above). Secondly, a hydrological model is calibrated and validated for the baseline period, and natural streamflow is simulated for the impacted period by forcing

Comment [mm25]: Sentence not useful

Comment [mm26]:

[Figure]

only meteorological data to a hydrological model. Therefore, this can be referred as the hydrological responses to climate variability only (Hu et al., 2015). The changes between observed streamflows for a baseline period and for an impacted period describe the combined impacts of increased human activities and climate variability (Wang et al., 2013;Hu et al., 2015), and this can be calculated as below:

$$\Delta = \Delta + \Delta = _{\boxed{}} -$$  12

where is the observed streamflow for a baseline period and $_{\boxed{}}$ for an impacted period. $\Delta$ demonstrates the total changes in streamflow, which is the sum of changes in streamflow due climate variability ($\Delta$) and changes in streamflow due to

10  human activities ($\Delta$). In the previous studies, such as (Wang et al., 2013;Dey and Mishra, 2017;Hu et al., 2015;Wang et al., 2008), $\Delta$ has been estimated by subtracting the average observed streamflow of baseline periods from average simulated streamflow of impacted periods. Since a model cannot simulate streamflow exactly the same as observed streamflow, and there are always some biases (e.g., model under or overestimates) between simulated and observed streamflows, which we can calculated during the model calibration and validation. For example, during the calibration and validation process in this study,

15  4–8% biases were obtained between simulated and observed streamflow (Table 5). This can mislead the original changes due to climate change and human activities. Therefore, in the present study, to reduce the effects of these biases, $\Delta$ was estimated by taking the difference between the simulated natural streamflow of the baseline period () and the natural streamflow of impacted period (), as below:

$$\Delta = -$$  13

After estimation of $\Delta$, the change in streamflow due to human activities ($\Delta$) can be obtained by taking the difference of total changes and changes due to climate, as in (Dey and Mishra, 2017), as below:

$$\Delta = \Delta - \Delta$$  14

 Finally, the percentage contributions of climate variability () and human activities () to total streamflow changes are quantified, as in (Dey and Mishra, 2017;Wang et al., 2008), as below:

$$(\%) = \frac{\Delta_{\boxed{}}}{\Delta_{\boxed{}}} \times 100 \text{ and } (\%) = \frac{\Delta_{\boxed{}}}{\Delta_{\boxed{}}} \times 100$$

In the present study, after the change point detection, HEC-HMS was calibrated and validated successfully for the baseline period (1951–1971), and then the natural streamflow was simulated by forcing the meteorological data for both the baseline period and the impacted period (1972–2013), at N'Djamena gauge in the CLRB. The simulated data from 1972–2015 was

**Comment [mm27]:** The present referee disagrees with this approach. Changes in rainfall may impact the natural vegetation, the groundwater level, the soil surface etc. All these can alter the rainfall-discharge relationship and create a change point.

[revised manuscript text omitted]

**Comment [mm36]:** If there is a change point somewhere, is it significant to draw a straight line for the whole period ? There is an obvious change point around 1988 ( not 1971)

[Figure]

[Figure]

**Figure 6.** Inflow ($m^3s^{-1}$) to Lake Chad from the Chari River, the El-Beid River, and the Komadugu-Yobe River along with the trend lines (Sen's slope estimates (SE)).

**Comment [mm37]:** Same comment for this figure

[Figure]

[Figure]

**Figure 7.** Change point year in streamflow at N'Dejamena gauge located on the Chari-Logone River detected by Worsley Likelihood Ratio Test (red line) and Standard Normal Homogeneity Test (black line).

[Figure]

[Figure]

[Figure]

[Figure]

[Figure]

[Figure]

**Figure 8.** Comparison of simulated and observed streamflow at N'djamena gauge for the (a) calibration period (1955–1965) and (b and c) validation periods (1966–1971 and 1951–1955) in the Chari-Logone River basin

[Figure]

[Figure]

[Figure]

**Figure 9.** Analyzing annual precipitation onf the basin along with naturalized (simulated) and observed flows at N'Djamena station in the Chari-Logone River basin.

**Comment [mm38]:** Consult LCBC where there is a hydrological model that fits more closely the Chari-Logone discharge

---

## Referee Comment (RC2) · Anonymous Referee #2 · 24 Apr 2018

In this manuscript the authors investigate the climatic variability and quantify the separate and combined impacts of human activities and climate change on the streamflow of Lake Chad basin from the period 1951-2015. They applied statistical trend tests and hydrological modeling. The results showed increasing trend in mean temperature, and decreasing signals in precipitation, with a decreasing trend of streamflow to Lake Chad. Furthermore, the impacts of human activities for the reduction of streamflow is more substantial than the impacts of climate variability. In general, the topic is scientifically challenging and is relevant for proper water resource management.

The differentiation between climate impact and human impact on the river discharge

into Lake Chad is performed in a rather simple way. A baseline period ("normal" climate before the detected breakpoint in 1971) is defined and the hydrological model is calibrated and validated for this period. For the remaining period up to present conditions the model is rum with the calibrated parameters. The deviation between the measured and the simulated hydrographs after the breakpoint is associated to human impact. This approach is intriguing, but has to be proved in a more rigorous way. Missing points are (see also my comments in the paper (pdf)): 1. Estimation of the uncertainty of the hydrological model (sensitivity of the parameters), 2. Interpolation and associated uncertainty of the meteorological input variables, 3. Cross checking of the results by incorporating the irrigation areas into the hydrological model.

The conclusion that water transfer from the Congo River is the best solution is not scientifically proven. There are many other options in the framework of Integrated Water Resources Management. The authors should skip this conclusion (It is not part of the paper) and write a second paper about it.

Further comments, which are not included directly into the pdf:

1. On page 5, L11-L12: monthly data of 11 meteorological (six for the period of 1950–2013 and other for 1985–2013) stations and 7 hydrometric stations (four for 1997–2007, two for 1951–2007, and one for 1951–2013) were collected from the Lake Chad Basin Commission (LCBC). However, on page 14, L6-L7, only the three stations of TM and PP were compared with CRU data for validation using statistical indicators. As the study area is very large, spatial variability is expected, and hence, validation of CRU data at three stations is not enough to capture the spatial variability.

2. On page L26-L28, the surface area of LC is decreased from 25, 0000 km 2 to 300 km2 in the 1980s. Moreover, the lake was divided into two parts in 1975 because of devastating drought over the African Sahal belt. This showed that climatic variability has a great impact on the hydrology of the Lake Chad. However, the findings of this paper is different (i.e. on the whole, an average decrease of 40% was estimated due

to climate variability and human activities for the period of 1972–2013, of which 66% of total decline was due to human activities and 34% due to climate). It is hardly possible to find a justification that can prove your model result. How do you explain this contradiction?

3. The potential impacts of irrigation projects are usually carried out during feasibility studies and detailed design of the irrigation fields. Please cite the outcome of these (governmental) studies and explain why the impact of the irrigation on discharge is higher than estimated.

4. Deficit and constant loss method that you used for your HEC-HMS model is referred to as event model in the HEC-HMS technical reference manual on page 40. This event model simulates behavior of the hydrologic system during a precipitation event while soil moisture accounting loss model is a continuous model that simulates both wet and dry weather behavior. So, base flow simulation during the dry weather might be questionable in your model?

5. HEC-HMS model is a lumped model in which spatial variations are averaged or ignored. Hence, the application of HEC-HMS for such large area (967,000 km2) considering the same landcover, soil type and other catchment characteristics might have an effect on the result.

6. Why human impact becoming dominant for the decreasing of the streamflow? How much water is extracted for irrigation will help to understand the implication of separate human activities as recommended by this manuscript. The role of evapotranspiration combination of both human and climate variability is also missing.

Minor comments a.. Description of study area is too much and there is a redundancy in different section of the manuscript which sometimes confused to understand. b. On page 2, L21....only in the last century (906-2015)... what is 906?? c. On page 2, L30... 1973-105... "105" may be error d. In the manuscript the word "streamflow"; " flow" and "runoff" used interchangeably. So, better to used one word consistently e. time

period for analysis is not consistent for instance, 1951-2015, 1951-2016, 1951-2013....
f. On page 6, L16 911000 km2 is mentioned which is different from 967000km2 g. On
page 6, L26-L27, for each subbasin, meteorological variables were obtained by taking
the average of all CRU grids covering that basin. How do you deal about the spatial
variability of the climate? why not used some interpolation techniques? h. Figure 3 is
not clear, needs improvement

Please also note the supplement to this comment:
https://www.hydrol-earth-syst-sci-discuss.net/hess-2018-139/hess-2018-139-RC2-
supplement.pdf
* * *
139, 2018.

**Supplement:**

[revised manuscript text omitted]

---

## Author Comment (AC1) · 9 May 2018

We would like to offer many thanks to the reviewer-1 (RC1) for his valuable comments. His comments will highly increase the quality of this manuscript. Responses to the RC1 has been attached here. There are three files for the RC1: 1) Response to the Reviewer 1: This describes general response to the RC1. 2) hess-2018-139-RC1-supplement: This include all responses to the specific comments from the RC1 3) HESS_Manuscript_May_2018_Revision1: This is the revised version of the manuscript according to the comments from both reviewers. The incorporated comments from the RC1 have been highlighted with green color, for easy follow up.

Please also note the supplement to this comment: https://www.hydrol-earth-syst-sci-discuss.net/hess-2018-139/hess-2018-139-AC1-supplement.zip

———————————————————

---

## Author Comment (AC2) · 9 May 2018

We would like to offer many thanks to the reviewer-2 (RC2) for his valuable comments. His comments will highly increase the quality of this manuscript. Responses to the RC1 has been attached here. There are three files for the RC2: 1) Response to the Reviewer_2: This describes general response to the RC2. 2) hess-2018-139-RC1-supplement: This include all responses to the specific comments from the RC2 3) HESS_Manuscript_May_2018_Revision1: This is the revised version of the manuscript according to the comments from both reviewers. The incorporated comments from the RC2 have been highlighted with yellow color, for easy follow up.

[Figure]

Please also note the supplement to this comment:
https://www.hydrol-earth-syst-sci-discuss.net/hess-2018-139/hess-2018-139-AC2-supplement.zip